# The phylogenetic landscape and nosocomial spread of the multidrug-resistant opportunist *Stenotrophomonas maltophilia*

Matthias I. Gröschel [1,2], Conor J. Meehan [3], Ivan Barilar [1], Margo Diricks [4], Aitor Gonzaga[5], Matthias Steglich[5], Oscar Conchillo-Solé [6,7], Isabell-Christin Scherer[8], Uwe Mamat[9], Christian F. Luz [10], Katrien De Bruyne[4], Christian Utpatel[1], Daniel Yero [6,7], Isidre Gibert [6,7], Xavier Daura [6,11], Stefanie Kampmeier[12], Nurdyana Abdul Rahman[13], Michael Kresken[14,15], Tjip S. van der Werf [2], Ifey Alio[16], Wolfgang R. Streit [16], Kai Zhou[17,18], Thomas Schwartz [19], John W.A. Rossen [10], Maha R. Farhat [20,21], Ulrich E. Schaible [9,22,23], Ulrich Nübel [5,23,24,25], Jan Rupp [8,22,28], Joerg Steinmann[26,27,28], Stefan Niemann[1,22,23,28 ✉] & Thomas A. Kohl [1,22,28]

Recent studies portend a rising global spread and adaptation of human- or healthcare-associated pathogens. Here, we analyse an international collection of the emerging, multi-drug-resistant, opportunistic pathogen *Stenotrophomonas maltophilia* from 22 countries to infer population structure and clonality at a global level. We show that the *S. maltophilia* complex is divided into 23 monophyletic lineages, most of which harbour strains of all degrees of human virulence. Lineage Sm6 comprises the highest rate of human-associated strains, linked to key virulence and resistance genes. Transmission analysis identifies potential outbreak events of genetically closely related strains isolated within days or weeks in the same hospitals.

[1] Molecular and Experimental Mycobacteriology, Research Center Borstel, Borstel, Germany. [2] Department of Pulmonary Diseases & Tuberculosis, University Medical Center Groningen, University of Groningen, Groningen, The Netherlands. [3] School of Chemistry and Bioscience, University of Bradford, Bradford, United Kingdom. [4] bioMérieux, Applied Maths NV, Keistraat 120, 9830 St-Martens-Latem, Belgium. [5] Leibniz Institute DSMZ - German Collection of Microorganisms and Cell Cultures, Braunschweig, Germany. [6] Institute of Biotechnology and Biomedicine, Universitat Autònoma de Barcelona, Barcelona, Spain. [7] Department of Genetics and Microbiology, Universitat Autònoma de Barcelona, Barcelona, Spain. [8] Department of Infectious Diseases and Microbiology, University Hospital Schleswig-Holstein, Lübeck, Germany. [9] Cellular Microbiology, Research Center Borstel, Borstel, Germany. [10] Department of Medical Microbiology and Infection Prevention, University Medical Center Groningen, University of Groningen, Groningen, The Netherlands. [11] Catalan Institution for Research and Advanced Studies, Barcelona, Spain. [12] Institute of Hygiene, University Hospital Münster, Münster, Germany. [13] Department of Microbiology, Singapore General Hospital, Singapore, Singapore. [14] Antiinfectives Intelligence GmbH, Rheinbach, Germany. [15] Rheinische Fachhochschule Köln gGmbH, Cologne, Germany. [16] Department of Microbiology and Biotechnology, University of Hamburg, Hamburg, Germany. [17] Shenzhen Institute of Respiratory Diseases, the First Affiliated Hospital (Shenzhen People's Hospital), Southern University of Science and Technology, Shenzhen, China. [18] Second Clinical Medical College, Jinan University, Shenzhen, China. [19] Karlsruhe Institute of Technology, Institute of Functional Interfaces, Eggenstein-Leopoldshafen, Germany. [20] Department of Biomedical Informatics, Harvard Medical School, Boston, MA, USA. [21] Division of Pulmonary and Critical Care, Massachusetts General Hospital, Boston, MA, USA. [22] German Center for Infection Research (DZIF), partner site Hamburg - Lübeck - Borstel - Riems, Cologne, Germany. [23] Leibniz Research Alliance INFECTIONS'21, Cologne, Germany. [24] Germany Center for Infection Research (DZIF), partner site Hannover - Braunschweig, Cologne, Germany. [25] Braunschweig Integrated Center of Systems Biology (BRICS), Technical University, Braunschweig, Germany. [26] Institute of Medical Microbiology, University Medical Center Essen, Essen, Germany. [27] Medical Microbiology and Infection Prevention, Institute of Clinical Hygiene, Paracelsus Medical Private University, Klinikum Nürnberg, Nuremberg, Germany. [28] These authors jointly supervised this work: Jan Rupp, Joerg Steinmann, Stefan Niemann, Thomas A. Kohl. ✉email: sniemann@fz-borstel.de

Recently, local transmission and global spread of hospital-acquired pathogens such as *Mycobacterium abscessus* and *Mycobacterium chimaera* were revealed by whole-genome sequencing (WGS), thereby challenging the prevailing concepts of disease acquisition and transmission of these pathogens in the hospital setting[1–3]. Global genome-based collections are missing for other emerging pathogens such as *Stenotrophomonas maltophilia*, listed by the World Health Organization as one of the leading drug-resistant nosocomial pathogens worldwide[4]. *S. maltophilia* is ubiquitously found in natural ecosystems, and is of importance in environmental remediation and industry[5,6]. *S. maltophilia* is an important cause of hospital-acquired drug-resistant infections with a significant attributable mortality rate in immunocompromised patients of up to 37.5%[7]. Patients under immunosuppressive treatment and those with malignancy or pre-existing inflammatory lung diseases such as cystic fibrosis are at particular risk of *S. maltophilia* infection[8]. Although almost any organ can be affected, mere colonisation needs to be discriminated from infections that mainly manifest as respiratory tract infections, bacteraemia or catheter-related bloodstream infections[5]. Yet, the bacterium is also commonly isolated from wounds and, at lower frequency, in implant-associated infections[9,10]. Furthermore, community acquired infections have also been described[11]. Treatment options are limited by resistance to a number of antimicrobial classes such as most β-lactam antibiotics, cephalosporins, aminoglycosides and macrolides through the intrinsic resistome, genetic material acquired by horizontal transfer, as well as non-heritable adaptive mechanisms[12,13].

To date, no large-scale genome-based studies on the population structure and clonality of *S. maltophilia* in relation to human disease have been conducted. Previous work indicated the presence of at least 13 lineages or species-like lineages in the *S. maltophilia* complex, defined as *S. maltophilia* strains with 16S rRNA gene sequence similarities >99.0%, with nine of these potentially human-associated[14–18]. These *S. maltophilia* complex lineages are further divided into four more distantly related lineages (Sgn1-4) and several *S. maltophilia* sensu lato and sensu stricto lineages[14,19]. The *S. maltophilia* strain K279a, isolated from a patient with bloodstream infection, serves as an indicator strain of the lineage *S. maltophilia* sensu stricto[19].

To understand the global population structure of the *S. maltophilia* complex and the potential for global and local spread of strains, in particular of human-associated lineages, we performed a large-scale genome-based phylogenetic and cluster analysis of a global collection of newly sequenced *S. maltophilia* strains together with publicly available whole-genome data.

## Results

**Strain collection and gene-by-gene analysis**. To allow for standardised WGS-based genotyping and gene-by-gene analysis of our data set, we first created an *S. maltophilia* complex whole-genome multilocus sequence typing (wgMLST) scheme. This approach, implemented as core genome MLST, has been widely used in tracing outbreaks and transmission events for a variety of bacterial species[20–22]. The use of a wgMLST scheme allows to analyse sequenced strains by their core and accessory genome[23]. Using 171 publicly available assembled genomes of the *S. maltophilia* complex that represent its currently known diversity (Supplementary Data 1), we constructed a wgMLST scheme consisting of 17,603 loci (Supplementary Data 2). To ensure compatibility with traditional MLST/gyrB typing methods, the wgMLST scheme includes the partial sequences of the seven genes used in traditional MLST as well as the gyrB gene[24] (Supplementary Table 1 and Supplementary Data 2).

To investigate the global phylogeographic distribution of *S. maltophilia*, we gathered WGS data of 2389 strains from 22 countries and four continents, which were either collected and sequenced in this study or had sequence data available in public repositories (Supplementary Data 3). All genome assemblies of the study collection passing quality thresholds (Supplementary Fig. 1, Supplementary Data 4) were analysed with the newly created wgMLST scheme. Upon duplicate removal, filtering for sequence quality and removal of strains with fewer than 2000 allele calls in the wgMLST scheme, our study collection comprised 1305 assembled genomes of majority clinical origin (87%) of which 234 were from public repositories and 1071 newly sequenced strains. Most strains came from Germany (932 strains), the United States (92 strains), Australia (56 strains), Switzerland (49 strains) and Spain (42 strains) (Fig. 1d; Supplementary Data 3). WgMLST analysis resulted in an average of 4174 (range 3024–4536) loci recovered per strain (Supplementary Data 5). Across the 1305 strains, most loci, 13,002 of 17,603, were assigned fewer than 50 different alleles (Supplementary Fig. 2). Calculation of the sample pan genome yielded 17,479 loci, with 2844 loci (16.3%) present in 95% and 1275 loci (7.3%) present in 99% of strains (Supplementary Fig. 3A). The pan genome at scale is not structured, likely due to extensive horizontal gene transfers[25] (Supplementary Fig. 3B). The genome sizes ranged from 4.04 Mb to 5.2 Mb.

**S. maltophilia complex comprises 23 monophyletic lineages**. To investigate the global diversity of the *S. maltophilia* complex, a maximum likelihood phylogeny was inferred from a concatenated sequence alignment of the 1275 core loci present in 99% of the 1305 *S. maltophilia* strains of our study collection (Fig. 1a). Hierarchical Bayesian analysis of population structure (BAPS), derived from the core single-nucleotide polymorphism (SNP) results, clustered the 1305 genomes into 23 monophyletic lineages named Sgn1–Sgn4 and Sm1–Sm18, comprising 17 previously suggested and six hitherto unknown lineages (Sm13–Sm18). For consistency, we used and amended the naming convention of lineages from previous reports[14,16]. In concordance with these studies[14,16], we found a clear separation of the more distantly related lineages Sgn1–Sgn4 and a branch formed by lineages Sm1–Sm18 (previously termed *S. maltophilia* sensu lato), with the largest lineage Sm6 (also known as *S. maltophilia* sensu stricto) containing most strains ($n = 413$), including the strain K279a and the species type strain ATCC 13637. Contrary to previous analyses, Sgn4 is the lineage most distantly related to the rest of the strains[14]. The division into the 23 lineages is also clearly supported by an average nucleotide identity (ANI) analysis (Fig. 1b, c). ANI comparisons of strains belonging to the same lineage was above 95%, and comparisons of strains between lineages were below 95%.

To evaluate structural genomic variation across the various lineages, we compiled a set of 20 completely closed genomes covering the 15 major phylogenetic lineages of both environmental and human-invasive or human-non-invasive isolation source. These genomes were either procured from the NCBI ($n = 8$) or newly sequenced on the PacBio platform ($n = 12$) (Supplementary Table 2). Interestingly, no plasmids were detected in any of the genomes. A genome-wide alignment of the 20 genomes demonstrated considerable variation in both structure and size between strains of different lineages and even strains of the same lineage (Supplementary Fig. 4). Several phage-related, integrative and conjugative mobile elements were observed across the genomes.

**Delineation of the S. maltophilia complex within its genus**. We calculated a phylogenetic tree based on an alignment of 23 predicted amino acid sequences from reference genes[15] to visualise the clade formed by strains of the *S. maltophilia* complex within

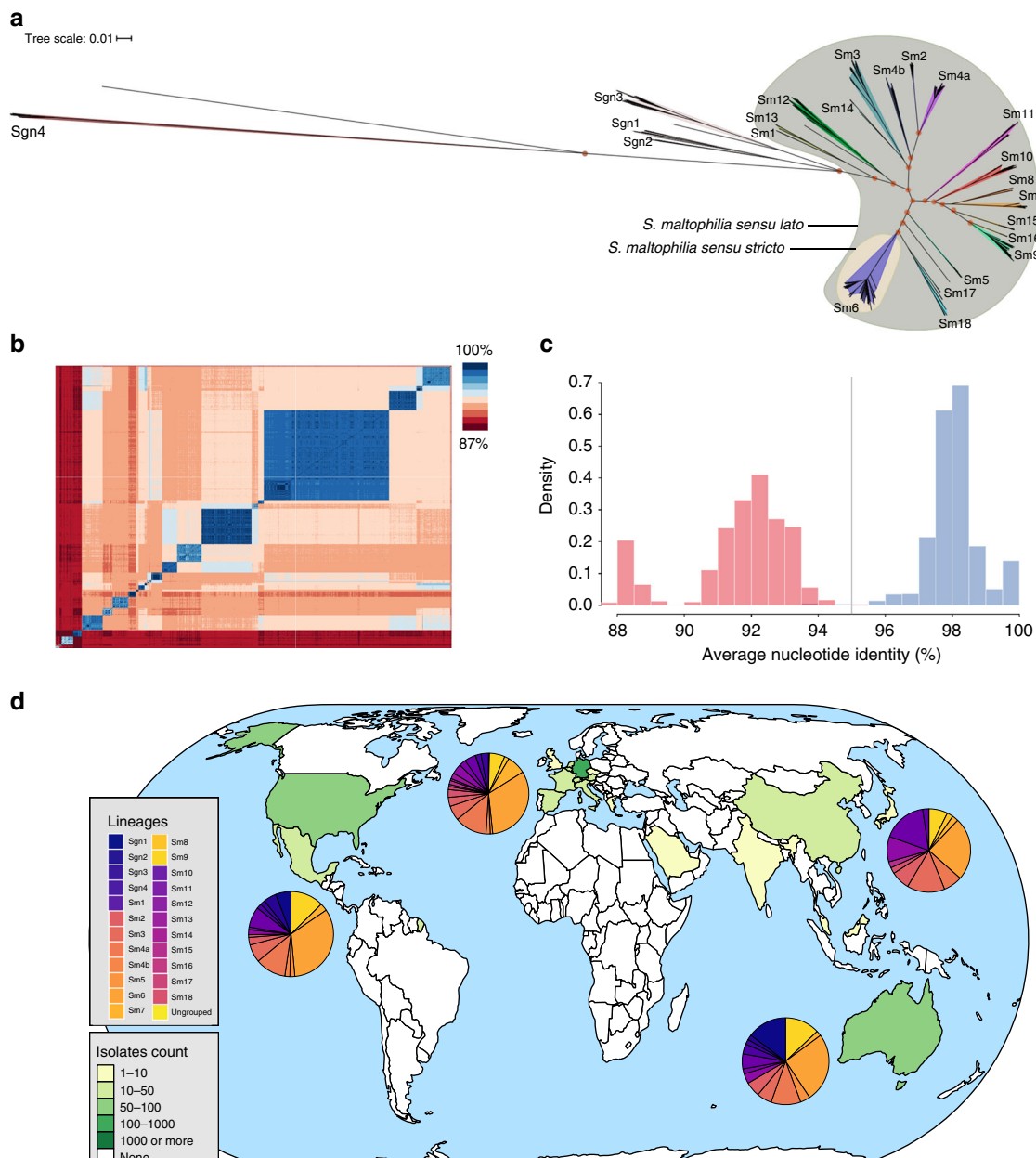

**Fig. 1 The global population structure of the *S. maltophilia* complex is composed of 23 monophyletic, globally distributed lineages.** **a** Unrooted maximum likelihood phylogenetic tree of 1305 *S. maltophilia* strains displaying the known population diversity of the *S. maltophilia* complex. The tree was built using RAxML on the sequences of 1275 concatenated core genome genes. Groups as defined by hierarchical Bayesian clustering are marked with shaded colours, and group numbers are indicated at the tree leafs of each corresponding group. orange shading = *S. maltophilia* sensu stricto; green shading = *S. maltophilia* sensu lato; 100% support values for the main branches are indicated with red circles. **b** Pairwise average nucleotide identity comparison calculated for 1305 *S. maltophilia* strains shown on a heatmap with blue indicating high and red indicating low nucleotide identity. **c** Histogram of pairwise average nucleotide identity (ANI) values, illustrating that strains of the same lineage are highly similar at the nucleotide level with ANI values above 95% (depicted in blue). Inter-lineage comparisons (in red colour) reveal low genetic identity between strains. The currently accepted species delimitation threshold at 95% is shown as a grey vertical line. **d** Geographic origin of the 1305 *S. maltophilia* strains comprising the study collection indicated on a global map. The green/yellow colour code indicates the number of strains obtained per country. The distribution of phylogenetic lineages per continent is displayed as colour-coded pie charts. The map was created using the tmap package in R[69]. Source data are provided as Source Data Files.

the genus *Stenotrophomonas* (Supplementary Fig. 5). When using the wgMLST scheme at the genus level, we recovered between 380 loci in *S. dokdonensis* and a maximum of 1677 in *S. rhizophila*, with *S. terrae*, *S. panacihumi*, *S. humi*, *S. chelatiphaga*, *S. daejeonensis*, *S. ginsengisoli*, *S. koreensis* and *S. acidaminiphilia* species receiving allele calls between these two values. For the strain *S. maltophilia* JCM9942 (Genbank accession GCA_001431585.1),

only 982 loci were detected. Interestingly, the 16S rRNA gene sequence of JCM9942 matched to that of *S. acidaminiphila*, and the JCM9942 16S rRNA sequence is only 97.3% identical and has an average nucleotide identity (ANI) of 82.9% with that of *S. maltophilia* ATCC 13637. We note that this strain has been reclassified as *S. pictorum* per 21/12/2019. In contrast, the number of recovered loci matches those of *S. maltophilia* strains for

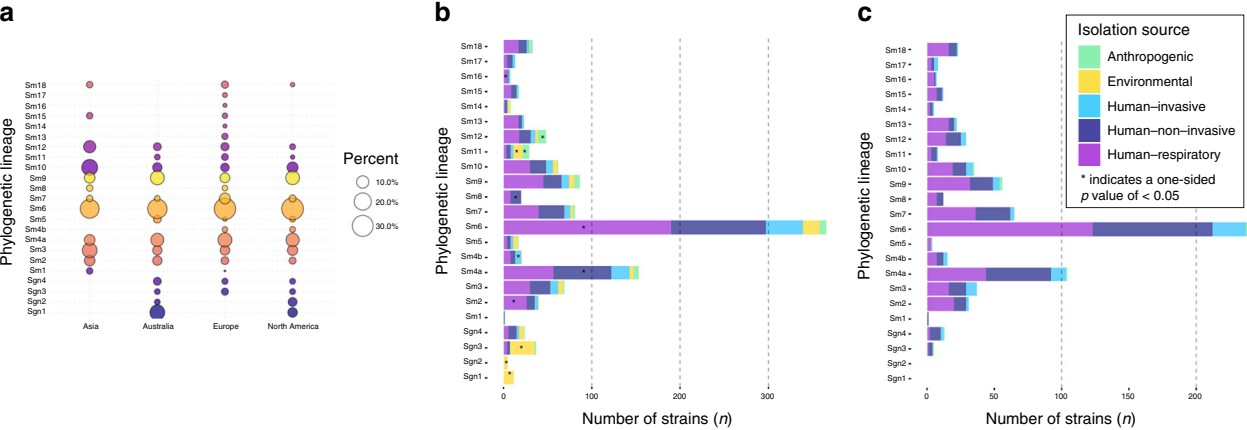

**Fig. 2 Global distribution of lineages, their composition by isolation source and contribution to phylogenetic lineage total number of strains. a** Bubble plot illustrating the proportion of lineages per continent. **b** Barplot showing the number of strains per lineage coloured by isolation source for the entire strain collection (see colour legend in **c**). **c** Barplot for the prospectively sampled representative collection of human-associated *S. maltophilia* complex strains per lineage coloured by human-invasive, human-non-invasive or human-respiratory. One-sided *p*-values for all within-lineage comparison of isolation sources can be found in Supplementary Table 3. $n = 1179$ *S. maltophilia* isolates where isolation source was known. '*' indicates a *p*-value of < 0.05 using one-sided Fisher's exact test (for $n < 5$) or test of equal and given proportions corrected for multiple testing using the Benjamini–Hochberg procedure.

*P. geniculata* (4060 loci), *S. lactitubi* (3805 loci), *S. pavanii* (3623 loci), and *S. indicatrix* (3678 loci). Here, 16S rRNA sequence comparison to *S. maltophilia* ATCC 13637 would support the inclusion of the first three of the aforementioned species with 16S rRNA sequence identity of >99.1% into the *S. maltophilia* complex[18,26].

**Lineages are globally distributed and differ in human association**. We next analysed the global distribution of strains of the lineages defined above and found that eight (Sm2, Sm3, Sm4a, Sm6, Sm7, Sm9, Sm10 and Sm12) are represented on all continents sampled within this study, with strains of lineage Sm6 accounting for the largest number of strains globally and the largest proportion on each sampled continent (Figs. 1d and 2a). To further investigate whether the lineages correlate with isolation source, particularly with regard to human host adaptation, we classified the isolation source of the *S. maltophilia* strains into five categories. Strains were considered environmental ($n = 117$) if found in natural environments, e.g. in the rhizosphere, and anthropogenic if swabbed in human surroundings such as patient room sink or sewage ($n = 52$). Human-invasive ($n = 133$) was used for isolates from blood, urine, drainage fluids, biopsies or in cerebrospinal fluid, human-non-invasive ($n = 353$) refers to colonising isolates from swabs of the skin, perineum, nose, oropharynx, wounds as well as intravascular catheters, and human-respiratory ($n = 524$) includes strains from the lower respiratory tract below the glottis and sputum collected from cystic fibrosis patients. For 126 strains, no information on their isolation source was available, and thus, these were not included in this analysis.

The more distantly related lineages Sgn1 (100%), Sgn2 (100%), Sgn3 (76%) and also Sm11 (38%) contained significantly more environmental strains compared with all other isolation sources in this lineage ($p < 0.001$, one-sided test of equal or given proportions or Fisher's exact test for $n < 5$, corrected for multiple testing using the Benjamini–Hochberg procedure), whereas strains of lineages Sm4a and Sm6 (3% and 5%, $p < 0.001$) were minority environmental (Fig. 2b, c; Supplementary Table 3). Anthropogenic strains were found at higher proportions in lineages Sm11 and Sm12 (24% and 19%, $p < 0.001$). Strains of lineage Sm4b were likely to be classified as human-invasive (33%, $p = 0.02$), and strains of lineage Sm4a and Sm8 were more likely to be human-non-invasive (42%, $p < 0.001$ and 60%, $p = 0.03$,

respectively). Sgn3 contained only few human-non-invasive strains (8%, $p = 0.02$). Strains of lineages Sm6 (52%, $p < 0.001$), Sm2 (65%, $p = 0.04$) and Sm13 (74%, $p = 0.03$) were linked to the human-respiratory isolation source. Strains of Sgn3 (11%, $p < 0.001$) and Sm11 (10%, $p < 0.001$) were less likely to be isolated from the human-respiratory tract (Fig. 2b).

The majority of strains sequenced within this study were prospectively collected through a hospital consortium across Germany, Austria and Switzerland ($n = 741$) (Fig. 2c). Restricting to this collection of human-associated strains from hospitalised patients, the most common lineage was Sm6 (33%) and lineages Sgn1-3 and Sm11, found to be environmentally associated in public data, were either not present or represented a minor proportion (<0.1%) of the collection.

We attempted genome-wide association (GWAS) to investigate the genetic correlates of human niche specificity using an elastic net whole-genome model that has recently been shown to outperform univariate approaches in controlling for population structure[27]. Using this approach, there was still considerable confounding due to residual population structure, likely related to the strong niche specificity of some of the lineages described above (Supplementary Fig. 8).

**Resistome and virulence characteristics of *S. maltophilia*.** We next screened our collection to detect potential resistance genes, e.g. chromosomally encoded antibiotic resistance genes, including efflux pumps[19,25,28]. We could identify members of the five major families of efflux transporters with high frequency in our strain collection (Fig. 3a)[19,29]. Aminoglycoside-modifying enzymes were encoded in 6.1% of strains (aminoglycoside-acetyltransferases) and 66% of strains (aminoglycoside-phosphotransferases), respectively, with five strains also harbouring aminoglycoside-nucleotidyltransferases. We observed that these enzyme families were unequally distributed among lineages, which preferentially contained either of the two major types. Strains of lineage Sm4a had the lowest proportion (4%, $p < 0.001$) of aminoglycoside-phosphotransferases. Taken together, 69% of the strains of our collection featured aminoglycoside-modifying enzymes. Other enzymes implicated in aminoglycoside resistance are the proteases ClpA and HtpX that were present in 99.3% and 99.8% of the strains investigated, respectively[30]. The *S. maltophilia* K279a genome encodes two

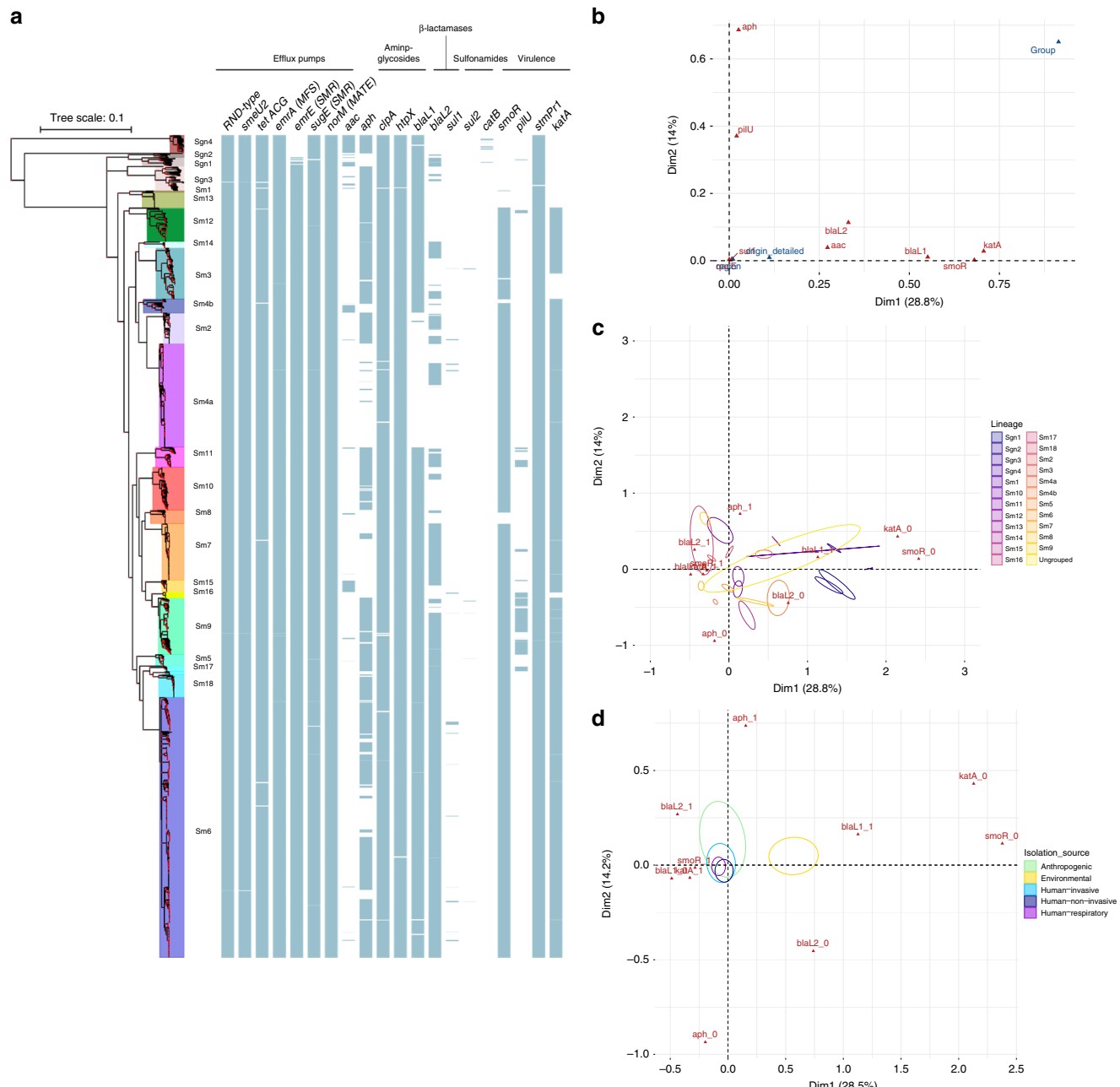

**Fig. 3 Resistance and virulence gene analysis. a** Midpoint rooted maximum likelihood phylogenetic tree based on 1275 core gene sequences of 1305 *S. maltophilia* complex strains. The coloured shading of the lineages represents the groups found by Bayesian clustering, with lineage names given. Hundred percent branch support is indicated by red dots. The pattern of gene presence (blue coloured line) or absence (white) is displayed in columns next to the tree, showing, from left to right, selected efflux pump genes: resistance-nodulation-cell-division (RND)-type efflux pumps, *smeU2* as part of the five-gene RND efflux pump operon *smeU1-V-W-U2-X*, *tetACG*, *emrA* of the major facilitator superfamily (MFS), *emrE* and *sugE* of the small-multidrug-resistance (SMR) efflux pump family, *norM* of the MATE family; the aminoglycoside acetyltransferase *aac* and phosphotransferase *aph*, *clpA*, *htpX*, the β-lactamases *blaL1* and *blaL2*, the *sul1* and *sul2* genes encoding dihydropteroate synthases, *catB*, and the virulence genes *smoR*, *pilU*, *stmPr1* and *katA*. **b** Variable correlation plot of a multiple correspondence analysis (MCA) visualising nine resistance and virulence genes as active variables in red and three supplementary variables region, origin and groupin blue. **c** Factor individual biplot map of phylogenetic lineages, indicated by their 99% confidence intervals (ellipses) across the first two MCA dimensions. The five highest contributing active variables are shown in red with 0 denoting absence and 1 presence of this variable. **d** Factor individual biplot map of the isolation source as indicated in the coloured legend. Source data are provided as Source Data file.

β-lactamases, the metallo-β-lactamase *blaL1* and the inducible Ambler class A β-lactamase *blaL2*[31]. While *blaL1* was found in 83.2% of our strains, *blaL2* was detected in only 63.2%. Interestingly, strains of some lineages lacked the *blaL2* gene, i.e. Sgn4, Sm1, Sm12, Sm13 and Sm16. Sm4a was the only lineage where no *blaL1* was found. Only one isolate encoded the oxacillin hydrolysing class D β-lactamase OXA. We noted a few strains harbouring the type B chloramphenicol-*O*-acetyltransferase CatB (0.6%). The sulfonamide-resistance-conferring *sul1* was seen in 19 strains (1.4%), and *sul2* was found in only 5 strains (0.4%), mostly occurring in human-associated or anthropogenic strains. This hints towards a low number of trimethoprim/sulfamethoxazole-resistant strains in our collection, which is the recommended first-line agent for the treatment of *S. maltophilia* infection[12].

We investigated the presence of virulence genes in our collection. SmoR is involved in quorum sensing and swarming motility of *S. maltophilia*, and was observed in 89.3% of our strains[32]. While SmoR was present in all Sm6 strains (proportion of 100%, $p < 0.001$, test of equal or given proportions, corrected for multiple testing using the Benjamini–Hochberg procedure), this gene was less prevalent in strains of lineages Sgn1 (7%, $p < 0.001$), and absent in strains of Sgn2, Sgn3 and Sgn4 (all $p < 0.001$). PilU, a nucleotide-binding protein that contributes to Type IV pilus function, was found in 9% of strains and mainly in lineages Sm9 (proportion of 81%, $p < 0.001$,) and Sm11 (proportion of 50%, $p < 0.001$)[33]. StmPr1 is a major extracellular protease of *S. maltophilia* and is present in 99.8% of strains[34]. KatA is a catalase mediating increased levels of persistence to hydrogen peroxide-based disinfectants and was found in 86.6% of strains[35]. While KatA is present at high proportions in strains of most lineages (i.e. Sm6 with 99% or Sm4a with 99%), Sm3 harbours this gene in only 49% ($p < 0.001$) of its strains, whereas it is absent in Sgn1, Sgn2, Sgn3 and Sgn4 (all $p < 0.001$). Taken together, *S. maltophilia* strains harbour a number of resistance-conferring as well as virulence genes, some of which are unequally distributed over the lineages.

We used multiple correspondence analysis (MCA) to investigate the correlation of the resistance and virulence profiles of the strains with geographic origin, isolation source and phylogenetic lineage. A total of nine genes, derived from virulence databases[36,37], that were either present or absent in at least ten isolates were selected to serve as active variables for the MCA (*aac, aph, blaL1, blaL2, katA, pilU, qacE, smoR, sul1*). As expected with a complex data set, the total variance explained by the MCA model was relatively low with the first four dimensions explaining 65.2% of variance in the model (Supplementary Fig. 6A). Nevertheless, from examining the first two dimensions of the MCA (accounting for 28.8% and 14% of variance), we noted that the genes *smoR, katA, blaL1, blaL2* and *aac* correlate with the first dimension of the MCA, while genes *aph* and *pilU* are corresponding to the second dimension (Fig. 3b; Supplementary Fig. 6B). When introducing geographic origin, isolation source and phylogenetic lineage as supplementary variables to the model, we observed a strong correlation of phylogenetic lineages with both dimensions, whereas little to no correlation was observed for isolation source and geographic origin (Fig. 3b). This indicates that virulence and resistance profiles of the nine genes are largely lineage-specific, with little impact of geographic origin or isolation source. However, we found a clear separation of the environmental strains from the rest of the collection when analysing the impact of human versus environmental habitat on the observed variance (Fig. 3d). A more detailed analysis of the observed lineage-specific variation, based on the explained variance from the MCA analysis, reveals that the more distantly related lineages Sgn1-4, Sm1 and Sm13 are characterised by the lack of *smoR, katA* and the presence of the aminoglycoside-acetyltransferases *aac*. The human-associated lineages Sm2, Sm6 and Sm7 are associated with the presence of *blaL2, sul1, blaL1, smoR* and *katA* (Fig. 3c).

**Possible local spread derived from genetic diversity analysis.** The identification of widely spread clonal complexes or potential outbreak events of *S. maltophilia* complex strains would have significant implications for preventive measures and infection control of *S. maltophilia* in clinical settings. We assessed our strain collection for circulating variants and clustered strains using the 1275 core genome MLST loci, that were also used for phylogenetic inference, and thresholds of 100 (d100 clusters) and 10 mismatched alleles (d10 clusters) for single-linkage clustering

(Fig. 4a). These thresholds were chosen based on the distribution of allelic mismatches (Fig. 4b). We found 765 (63%) strains to group into 82 clusters (median cluster size 6, IQR 6–11.7) within 100 alleles difference. A total of 270 (21%) strains were grouped into 62 clusters within 10 alleles difference (median cluster size 4, IQR 3–4.7). The maximum number of strains per cluster were 45 and 12 for the d100 and d10 clusters, respectively. Interestingly, strains within d100 clonal complexes originated from different countries or cities (Fig. 5a).

Some strains of lineages, notably those with primarily environmental strains, did not cluster at d10 level at all (Sgn1-4, Sm1, Sm15) (Supplementary Table 4). The d10 clustering rate ranged from 18% for strains of lineage Sm4a to 48% for Sm13, while 21% of lineage Sm6 strains were in d10 clusters. When day and location of isolation were known and included in further investigations of the d10 clusters, we detected a total of 49 strains, grouped into 13 clusters (of at least two isolates), which were isolated from the same respective hospital in the same year. Of these, three d10 clusters consisted of strains isolated from the respiratory tract of different patients treated in the same hospital within an 8-week time span or less (Table 1, Fig. 5b; Supplementary Fig. 7).

## Discussion

The findings of this study demonstrate that strains of the human opportunistic pathogen *S. maltophilia* can be subdivided into 23 monophyletic lineages, with two of these comprising exclusively environmental strains. The remaining lineages contain strains from mixed environmental and human sources. Among these strains, certain lineages such as Sm6 are most frequently found to be human-invasive, human-non-invasive or human-respiratory strains, pointing towards a potential adaptation to human infection and enhanced virulence. This is supported by their association with antibiotic resistance genes, resulting in the multidrug resistance observed among human-associated lineages. Our data provide evidence for the global prevalence of particular circulating lineages with hospital-linked clusters collected within short time intervals suggesting transmission. The latter emphasises the need to instate or re-enforce hygiene and infection control practices to minimise in-hospital spread of these pathogens.

In line with previous reports, our large genome-based study revealed that the *S. maltophilia* complex is extraordinarily diverse at the nucleotide level, representing a challenge for population-wide analyses and molecular epidemiology[14–17]. To address this, we first developed a new genome-wide gene-by-gene typing scheme, consisting of 17,603 gene targets, or loci. This whole-genome MLST typing scheme provides a versatile tool for genome-based analysis of *S. maltophilia* complex strains and a unified nomenclature to facilitate further research on the complex with an integrative genotyping tool and sequence data analysis approach. Including the loci of the 7-gene classical MLST typing scheme as well as the *gyrB* gene enables backward compatibility and comparison of allele numbers with sequence types obtained through the classical MLST scheme[24]. Applying the wgMLST approach to our extensive and geographically diverse collection of *S. maltophilia* strains allowed us to infer a comprehensive phylogenetic population structure of the *S. maltophilia* complex, including the discovery of six previously unknown lineages in addition to those described previously[14,17].

Altogether, we found 23 distinct phylogenetic lineages of the *S. maltophilia* complex, which are well supported by hierarchical Bayesian clustering analysis of the core genome and intra- and inter-lineage average nucleotide identity. This genetic heterogeneity observed between the detected lineages is sufficient to

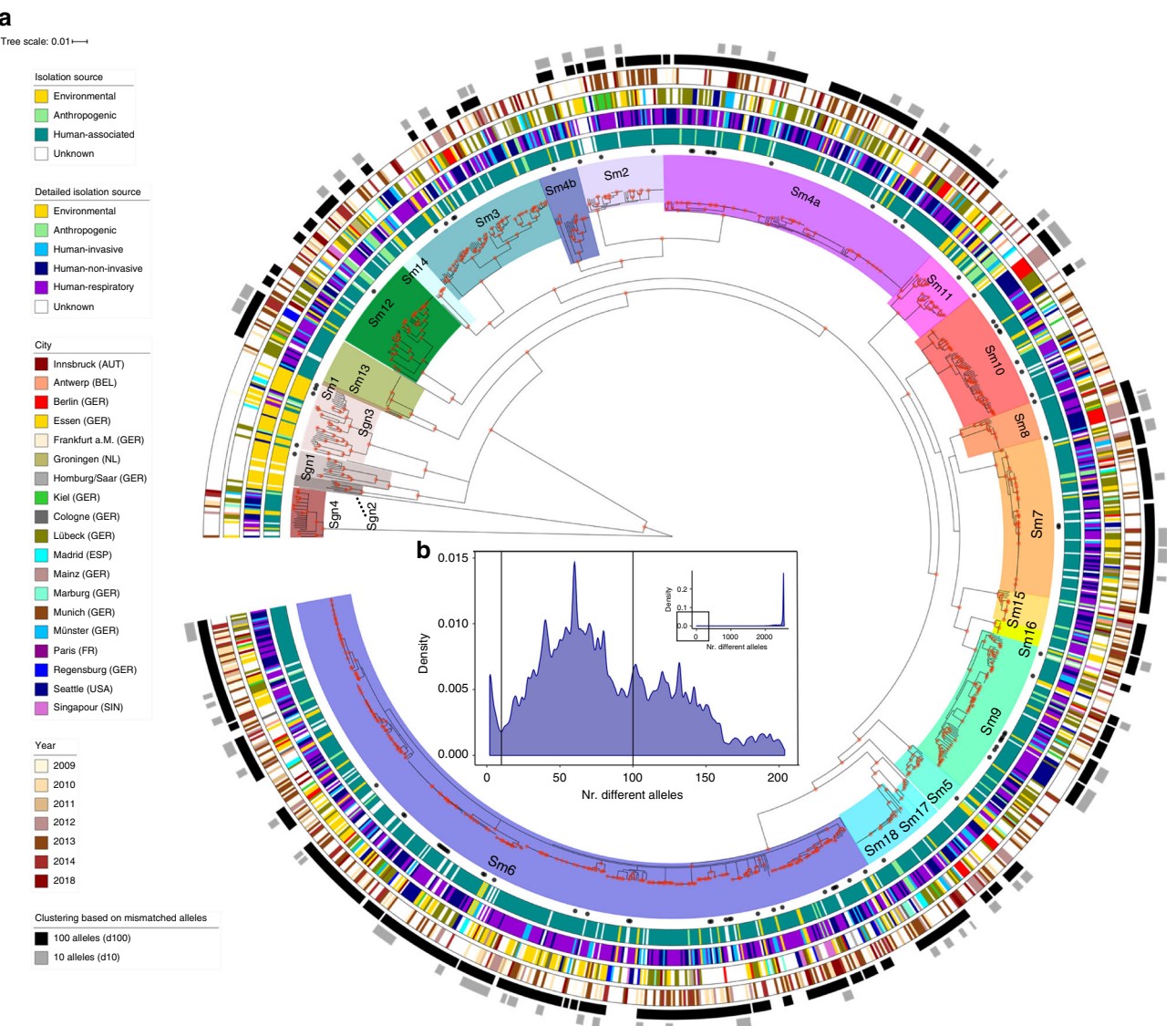

**Fig. 4 Spatiotemporal cluster analysis of 1305 S. maltophilia complex strains. a** The coloured ranges across the outer nodes and branches indicate the 23 lineages. The black dots indicate the location of the genome data sets used for wgMLST scheme generation. The rings, from inside towards outside denote (i) the isolation source of the strains classified as either environmental, anthropogenic, human or unknown; (ii) the detailed isolation source of strains similar to the first ring with the human strains subclassified into human-invasive, human-non-invasive and human-respiratory; (iii) the city of isolation; (iv) the year of isolation (where available), with light colours representing earlier years and darker brown colours more recent isolation dates. The outer rings in black-to-grey indicate the single-linkage-derived clusters based on the number of allelic differences between any two strains for 100 (d100 clusters) and 10 (d10 clusters) allelic mismatches. Red dots on the nodes indicate support values of 100%. **b** Distribution of the number of wgMLST allelic differences between pairs of strains among the 1305 S. maltophilia strains. The main figure shows the frequencies of up to 200 allelic differences, while the inset displays frequencies of all allelic mismatches. Source data are provided as Source Data Files.

consider them as clearly separate lineages of the *S. maltophilia* complex, in line with previous results from classical typing methods and phylogenetic studies[14–17,38]. In fact, the average nucleotide identity between lineages was below the threshold generally considered to define a species, warranting further studies on and possible revisions of the taxonomic assignments and nomenclature for this group. In parallel with these reports, human adaptation is observed to vary, with strains from lineages Sgn1, Sgn2, Sgn3 and Sm11 mostly isolated from the environment and strains from the other lineages mostly derived from human or human-associated sources. Apart from the purely environmental lineages Sgn1 and Sgn2, our results indicate that strains from all other lineages are able to colonise humans and cause infection, including lineage Sgn4 outside the "*sensu lato*" group, and

potentially switch back and forth between surviving in the environment and within a human host. These results do not support the notion that the *S. maltophilia sensu stricto* strains of lineage Sm6 represent the primary human pathogens[14]. We therefore propose to continue using the term *S. maltophilia* complex and the respective lineage classification for all strains that are identified as *S. maltophilia* by routine microbiological diagnostic procedures in hospitals and omit the use of *sensu stricto* or *lato*.

Beyond associating some of the lineages with either environment or human, we were not able to identify the specific genetic mechanisms that underlie this association due to the extent of stratification by population structure. *S. maltophilia* is believed to be a much less virulent pathogen relative to other nosocomials such as *P. aeruginosa* or *S. aureus*[39]. The establishment of human

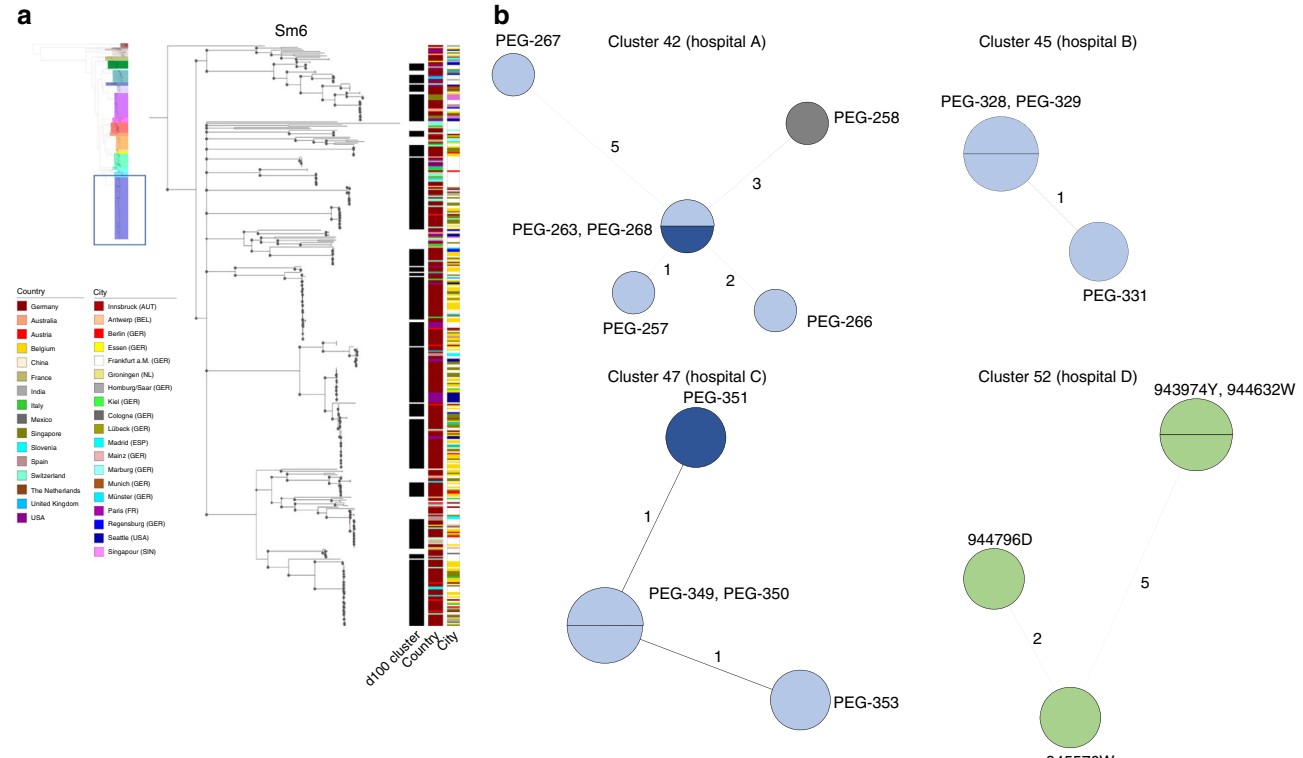

**Fig. 5 Analysis of d100 clusters in lineage Sm6 and closely related d10 clusters across the study collection. a** The d100 clusters in the largest human-associated lineage Sm6 consist of strains from various countries and, for strains from the same country, of various cities. The coloured bars represent, from left to right, the d100 clonal complexes, the country of isolation, and the city of isolation. **b** High-resolution analysis of four selected d10 allele clusters for which detailed metadata, i.e. day, source and ward of isolation, was available are shown as minimum spanning trees based on the 100% core genome MLST loci of the respective cluster. The number of loci used were 3734 for cluster 42 (hospital A), 4190 for cluster 45 (hospital B), 3637 for cluster 47 (hospital C) and 3714 for cluster 52 (hospital D). The number of mismatched alleles are shown in small numbers on the connecting branches. Node colours indicate isolation source, light blue = respiratory sample, dark blue = sputum, grey = wound swap, green = endoscope. Source data are provided as Source Data Files.

**Table 1 Site and date of isolation for the strains comprising the four d10 clusters isolated from the same geographic location within at most an 8-week time span.**

| Strain | Lineage | Cluster | Isolation date | Isolation place | Clinical source |
|---|---|---|---|---|---|
| PEG-257 | Sm2 | 42 | October 4th, 2013 | Hospital A | Respiratory tract |
| PEG-258 | | | October 7th, 2013 | | Wound swap |
| PEG-263 | | | October 9th, 2013 | | Respiratory tract |
| PEG-266 | | | October 14th, 2013 | | Respiratory tract |
| PEG-267 | | | October 14th, 2013 | | Respiratory tract |
| PEG-268 | | | October 14th, 2013 | | Sputum |
| PEG-328 | Sm18 | 45 | October 21st, 2013 | Hospital B | Respiratory tract |
| PEG-329 | | | October 21st, 2013 | | |
| PEG-331 | | | December 9th, 2013 | | |
| PEG-351 | Sm13 | 47 | January 11th, 2014 | Hospital C | Sputum |
| PEG-349 | | | January 21st, 2014 | | Respiratory tract |
| PEG-350 | | | January 24th, 2014 | | Respiratory tract |
| PEG-353 | | | January 27th, 2014 | | Respiratory tract |
| 943974Y | Sm12 | 52 | January 28th, 2014 | Hospital D | Endoscope |
| 944632W | | | February 17th, 2014 | | |
| 944796D | | | February 21st, 2014 | | |
| 945570W | | | March 24th, 2014 | | |

infection or colonisation with *S. maltophilia* is likely also strongly driven by the host factors such as the immune status while the role of pathogen genetic background or specific virulence mechanisms is still to be determined[40]. Collecting data on the host immune status or other predisposing factors will enable research in this area in the future.

Our results further illustrate that strains of nearly all 23 lineages are present in sampled countries and continents,

suggesting a long evolutionary trajectory of *S. maltophilia*. The finding that the more distantly placed lineages (Sgn1-4) as well as the other species of the genus *Stenotrophomonas* comprise primarily environmental strains lends to speculations that this trajectory took place from an exclusively environmental lifestyle towards human colonisation and infection. This could be due to the emergence of individual strains adapted to survive in both niches or to multiple, independent events of pathoadaptation of environmental strains to human colonisation, as has been observed for *Legionella pneumophila*[41]. A more recent study expanded these findings on the entire *Legionella* genus, illustrating that the capacity to infect eukaryotic cells can be acquired independently many times[42]. The evolution within the *S. maltophilia* complex might have been aided by the apparent genomic plasticity as seen from quite distinct genome lengths and structural variation, even within individual lineages. In addition, multiple pathoadaption events along with extensive horizontal gene transfer events could constitute one of the causes for the relatively large and non-structured accessory genome we detected[25]. A striking observation achieved by long-read PacBio sequencing was the absence of plasmids in the completed genomes that hence did not play a role in gene exchange and resistance development in the selected *S. maltophilia* strains.

It is well established that *S. maltophilia* is equipped with an armamentarium of antimicrobial resistance-conferring mechanisms[5,19]. In our strain collection, we found several families of antibiotic efflux pumps ubiquitously present among strains of all 23 lineages, as well as other genes implicated in aminoglycoside or fluoroquinolone resistance. In some cases, resistance-related genes were only present in some lineages, such as the β-lactamase gene *blaL2* or the aminoglycoside acetyl- and phosphotransferases genes *aac* and *aph*. Interestingly, those lineages harbouring mostly environmental strains tended to harbour less resistance and virulence genes than lineages that comprised at a majority human-associated strains. For instance, the four lineages most distantly placed from the remaining *S. maltophilia* complex, Sgn1–Sgn4, were associated with the lack of key virulence and resistance factors. In contrast, the human-associated lineage Sm6 was linked to the presence of a β-lactamase (BlaL2) and KatA, involved in resistance to disinfectants, pointing towards adaptation to healthcare settings and survival on and in patients. While other human-associated lineages also harboured resistance and virulence genes at high proportions, this finding might explain why strains of lineage Sm6 were dominant in our investigation, both in our total study collection as well as in the subset of prospectively collected strains as the majority of strains were isolated from human-associated sources. This notion is also supported by our finding that we did not detect any d100 clusters, or circulating variants, in the primarily environmental-associated lineages. Yet, in light of the low number of strains belonging to these lineages in our data set as well as the lack of systematic sampling for environmental isolates, these results should be interpreted with caution.

Importantly, our study indicates the presence of potential transmission clusters in human-associated strains, suggesting potential direct or indirect human-to-human transmission[17]. Indeed, we identified a remarkable number of closely related strains (270) that congregated in 62 clusters as indicated by a maximum of ten mismatched alleles in the pairwise comparison. While no d10 clusters were found in the more distantly placed lineages Sgn1-4, all other lineages comprised of such clusters with similar clustering rates. A common source of infection is supported in those cases where detailed epidemiological information concerning hospital and day of isolation was available. Further studies looking into potential transmission events are warranted as this would have major consequences on how infection

prevention and control teams deal with *S. maltophilia* colonisation or infection.

We are aware that our study is limited by our collection framework. Molecular surveillance of *S. maltophilia* is currently not routinely performed and no robust data on prevalence, sequence types or resistance profiles exist. The geographic restriction of our prospective sampling is biased towards the acquisition of clinical and human-pathogenic *S. maltophilia* strains from a multi-national consortium that mainly comprised German, Austrian and Swiss hospitals. The inclusion of all available sequence data in public repositories compensates this restriction partially, however, for these strains information on isolation source and date was incomplete or missing. More prospective, geographically diverse sampling from different habitats is warranted to corroborate our findings, especially concerning the apparent adaptation to the human host. Ultimately, it will be highly interesting to correlate genotype to patient outcomes to identify genomic groups that might be associated with a higher virulence.

Taken together, our data show that strains from several diverse *S. maltophilia* complex lineages are associated with the hospital setting and human-associated infections, with lineage Sm6 strains potentially best adapted to colonise or infect humans. Strains of this lineage are isolated worldwide, are found in potential human-to-human transmission clusters and are predicted to be highly resistant to antibiotics and disinfectants. Accordingly, strict compliance to infection prevention measures is important to prevent and control nosocomial transmissions especially of *S. maltophilia* lineage Sm6 strains, including the need to ensure that the commonly used disinfectants are effective against *S. maltophilia* complex strains expressing KatA. Future anti-infective treatment strategies may be based on our finding of a very low prevalence of trimethoprim-sulfomethoxazole-resistance genes in our collection, suggesting that this antibiotic drug remains the drug of choice for the treatment of *S. maltophilia* complex infections.

## Methods

**Bacterial strains and DNA isolation**. All *Stenotrophomonas maltophilia* complex strains sequenced in this study were routinely collected in the participating hospitals and identified as *S. maltophilia* using MALDI-TOF MS. The strains were grown at 37 °C or 30 °C in either lysogeny broth (LB) or Brain Heart Infusion media. RNA-free genomic DNA was isolated from 1-ml overnight cultures using the DNeasy Blood & Tissue Kit according to the manufacturer's instructions (Qiagen, Hilden, Germany). To ensure correct identification as *S. maltophilia*, the 16S rRNA sequence of *S. maltophilia* ATCC 13637 was blasted against all strains. The large majority (1278 strains, 98%) of our data set had 16S rRNA similarity values ≥ 99% (rounded to one decimal). Twenty-seven strains, mostly from the more distant clades Sgn1-4, had 16S rRNA blast results between 98.8% and 98.9%. Where no 16S rRNA sequence was found (one study using metagenome assembled genomes[43] as well as accession numbers GCA_000455625.1 and GCA_000455685.1) we left the isolates in our collection if the allele calls were above the allele threshold of 2000 (Supplementary Fig. 1H).

**Whole-genome data collection and sequencing**. We retrieved available *S. maltophilia* sequence read data sets and assembled genomes from NCBI nucleotide databases as of April 2018, excluding next-generation sequencing (NGS) data from non-Illumina platforms and data sets from studies that exclusively described mutants. For studies investigating serial strains from the same patient, we chose only representative strains, i.e. one sample per patient was chosen from Esposito et al.[44] and one strain of the main lineages found by Chung et al.[45]. In case of studies providing both NGS data and assembled genomes, we included the NGS data in our analysis.

In addition, we sequenced the genomes of 1071 clinical and environmental strains. NGS libraries were constructed from genomic DNA using a modified Illumina Nextera protocol[46] and the Illumina NextSeq 500 platform with 2 × 151 bp runs (Illumina, San Diego, CA, USA). NGS data were assembled de novo using SPAdes (v3.7.1) included into the BioNumerics software (v7.6.3, Applied Maths NV). We excluded assemblies with an average coverage depth < 30 × (Supplementary Fig. 1A), deviating genome lengths (< 4 Mb and > 6 Mb) (Supplementary Fig. 1B), number of contigs > 500 (Supplementary Fig. 1C), >2000 non-ACTG bases (Supplementary Fig. 1D), an average quality < 30 (Supplementary Fig. 1E) and GC content (< 63% or > 68%) (Supplementary

Fig. 1F). Fifty-five data sets where assembly completely failed were excluded from further analysis. For the phylogenetic analysis, we further excluded strains possessing <2000 genes of the whole-genome MLST scheme constructed in this study (Supplementary Fig. 1H). The resulting data set contained 1305 samples (234 from public databases) with a mean coverage depth of $130 \times$ (SD = 58; median 122, IQR 92–152), consisted of, on average, 74 contigs (mean, SD = 44; median 67, IQR 47–93) and encompassed a mean length of 4.7 million base pairs (SD = 0.19; median 4.76, IQR 4.64–4.87) (Supplementary Data 4). All assemblies were assessed for completeness (range 81.03–100, mean 99.7, SD = 1.3) and contamination (range 0–10.8, mean 0.38, SD = 0.59) using CheckM[47].

Next-generation sequencing data generated in the study are available from public repositories under the study accession number "PRJEB32355" (accession numbers for all data sets used are provided in Supplementary Data 3).

**Generation of full genomes by PacBio sequencing.** We used PacBio long-read sequencing on an RSII instrument (Pacific Biosciences, Menlo Park, CA, USA) to generate fully closed reference genome sequences of *S. maltophilia* complex strains sm454, sm-RA9, Sm53, ICU331, SKK55, U5, PEG-141, PEG-42, PEG-173, PEG-68, PEG-305 and PEG-390, which together with available full genomes, represent the majority of the diversity of our collection. SMRTbellTM template library was prepared according to the Procedure & Checklist—20 kb Template Preparation using the BluePippinTM Size-Selection System (Pacific Biosciences, Menlo Park, CA, USA). Briefly, for preparation of 15-kb libraries, 8 µg of genomic DNA from *S. maltophilia* strains was sheared using g-tubesTM (Covaris, Woburn, MA, USA) according to the manufacturer's instructions. DNA was end-repaired and ligated overnight to hairpin adapters applying components from the DNA/Polymerase Binding Kit P6 (Pacific Biosciences, Menlo Park, CA, USA). BluePippinTM Size-Selection to 7000 kb was performed as instructed (Sage Science, Beverly, MA, USA). Conditions for annealing of sequencing primers and binding of polymerase to purified SMRTbellTM template were assessed with the Calculator in RS Remote (Pacific Biosciences, Menlo Park, CA, USA). SMRT sequencing was carried out on the PacBio RSII (Pacific Biosciences, Menlo Park, CA, USA) taking one 240-minutes movie for each SMRT cell. In total, one SMRT cell for each of the strains was run. For each of the 12 genomes, 59,220–106,322 PacBio reads with mean read lengths of 7678–13,952 base pairs (bp) were assembled using the RS_HGAP_Assembly.3 protocol implemented in SMRT Portal version 2.3.0[48]. Subsequently, Illumina reads were mapped onto the assembled sequence contigs using BWA (version 0.7.12)[49] to improve the sequence quality to 99.9999% consensus accuracy. The assembled reads were subsequently disassembled for removal of low-quality bases. The contigs were then analysed for their synteny to detect overlaps between its start of the anterior and the end of the posterior part to circularise the contigs. Finally, the *dnaA* open-reading frame was identified and shifted to the start of the sequence. To evaluate structural variation, genomes were aligned using blastn. PlasmidFinder[50] was used to screen the completed genomes for plasmids. Genome sequences are available under bioproject number; the accession numbers can be found in Supplementary Table 2.

**Construction of a whole-genome MLST scheme.** A whole-genome multilocus sequence typing (wgMLST) scheme was created by Applied Maths NV (bioMérieux) using 171 publically available *S. maltophilia* genome data sets. First, an initial set of loci was determined using the coding sequences (CDS) of the 171 genomes (Supplementary Data S1). Within this set, loci that overlapped >75% or that yielded BLAST hits at the same position within one genome were omitted or merged until only mutually exclusive loci were retained while preserving maximal genome coverage. Mutually exclusive loci are defined as loci for which the reference alleles (typically one or two unique DNA sequences per loci) only yield blast hits at a threshold of 80% similarity to their own genomic location and not to reference alleles of another locus, such as paralogs or repetitive regions. In addition, loci that had a high rate of invalid allele calls (e.g. because of the absence of a valid start/stop codon [ATG, CTG, TTG, GTG], the presence of an internal stop codon [TAG, TAA, TGA] or non-ACTG bases) and loci for which alleles were found containing large tandem repeat areas were removed. Lastly, multi-copy loci, i.e. repeated loci for which multiple allele calls were retrieved, were eliminated to achieve 90% of the genome data sets used for scheme validation had <10 repeated loci. The resulting scheme contained 17,603 loci (including the seven loci from the previously published MLST scheme[24], see Supplementary Table 1) (Supplementary Figs. 2 and 3) and can be accessed through a plugin in the BioNumericsTM Software (Applied Maths NV, bioMérieux). On average, 4174 loci (range 3024–4536) were identified per genome of our study collection.

To determine the allele number(s) corresponding to a unique allele sequence for each locus present in the genome of a strain, two different algorithms were employed: the assembly-free (AF) allele calling uses a k-mer approach (k-mers size of 35 with minimum coverage of 3) starting from the raw sequence reads while the assembly-based (AB) allele calling performs a blastn search against assembled genomes with the reference alleles of each loci as query sequences. The word size for the gapped blast search was set at 11, and only hits with a minimum homology of 80% were retained. After each round of allele identification, all the available data from the two algorithms (AF and AB) were combined into a single set of allele assignments, called consensus calls. If both algorithms returned one or multiple allele calls for a given loci, the consensus is defined as the allele(s) that both analyses have in common. If there is no overlap, there will be no allele number

assigned for this particular locus. If for a specific locus the allele call is only available for one algorithm, this allele call will be included. If multiple allele sequences were found for a consensus locus, only the lowest allele number is retained. Genes of which the sequence was not yet in the allele database were only assigned an allele number in case the sequence had valid start/stop codons, had no ambiguous bases or internal stop codons, had at least 80% homology towards one of the reference allele sequences and had no more than 999 gaps in the pairwise sequence alignment towards the closest allele sequence from the same locus. The loci of the scheme were annotated using the blast2go tool[51] relying on NCBI blast version 2.4.0 + [52] and InterProscan 5 online[53] (Supplementary Data 2), and the November 2018 GO[54,55] and NCBI nr databases were used. All loci of the wgMLST scheme in FASTA format can be accessed using this link (https://figshare.com/articles/Smaltophilia_wgMLST_all-alleles_fasta_gz/10005047).

**Whole-genome MLST scheme validation.** To validate the scheme, publicly available sequence read sets from different publications[44,56] were analysed with the wgMLST scheme in BioNumerics (v7.6.3). In addition, wgMLST analysis was performed three times on the same sequence read set of two samples[56]. These technical replicates had the same number of consensus allele calls and the allele numbers were identical. The allelic profiles of three between-run replicates (sequencing data obtained from different fresh cultures) and three within-run replicates (sequencing data obtained from different libraries made using the same DNA extract of one fresh culture) of *S. maltophilia* strain ATCC 13637[56] were identical, except for one locus (STENO00008). The difference in allele calling for this locus, a gene coding for a ferric siderophore transport system/periplasmic transport protein tonB, is likely due to sequencing and assembly difficulties of this GC-rich gene. Replicating the core genome SNP tree from Esposito and colleagues[44] based on wgMLST results yielded a highly similar tree topology clustering samples from each patient with few exceptions.

**Phylogenetic analysis.** We characterised the core loci present in 99% of the data set based on loci presence, i.e. that genes received a valid allele call, amounting to 1275 loci. For phylogenetic analyses, a concatenated alignment of the 1275 core genes from all strains was created, and an initial tree was built using RAxML-NG with a GTR + Gamma model,using the site-repeat optimisation, and 100 bootstrap replicates[57]. This alignment and the tree were then fed to ClonalFrameML to detect any regions of recombination[58]. These regions were then masked using maskrc-svg (https://github.com/kwongj/maskrc-svg), and this masked alignment was then used to build a recombination-free phylogeny using the same approach as above in RAxML-NG. iTOL was employed for annotating the tree[59]. The core gene alignment length was 1,070,730 variants, amounting to 1,397,302,650 characters for the entire data set. Across all isolates, 593,506,119 positions (42% of all variants) were masked for recombination. For phylogenetic and BAPS analysis, all invariant sites were removed to obtain the final alignment length of 296,491 variants. The assemblies were annotated with prokka[60], and the pan genome was calculated and visualised using roary[61].

The genus wide tree showing a comparative phylogenetic analysis of the lineages with *Stenotrophomonas* species data (Supplementary Fig. 5) was calculated using IQtree[62] based on an alignment of the concatenated predicted protein sequence of 23 genes[15] (*dnaG, rplA, rplB, rplC, rplD, rplE, rplF, rplK, rplL, rplM, rplN, rplP, rplS, rpmA, rpoB, rpsB, rpsC, rpsE, rpsJ, rpsK, rpsM, rpsS, tsf*) that were extracted from the assemblies using blastn.

We detected phylogenetic lineages within the tree using a hierarchical Bayesian Analysis of Population Structure (hierBAPS) model as implemented in R (rHierBAPs) with a maximum depth of 2 and maximum population number of 100[63]. FastANI[64] was employed to calculate the pairwise average nucleotide identity (ANI) as a similarity matrix between all the strains with the option 'many-to-many'. The similarity matrix was imported into R and used together with the group assignment obtained from hierBAPS to compare the ANI values in strains within and between groups. ANI values were plotted as a heatmap of all strains as well as a composite histogram of identity between and within groups.

**Resistome and virulence analysis.** Resistome and virulome were characterised with abricate version 0.8.7[36] screened against the NCBI Bacterial Antimicrobial Resistance Reference Gene Database (NCBI BARRGD, PRJNA313047) and the Virulence Factors of Pathogenic Bacteria Database (VFDB)[37]. All genes below 80% coverage breadth were excluded. In addition, literature was reviewed to identify additional genes associated with antibiotic resistance, and virulence in *S. maltophilia* and the corresponding loci were extracted from the wgMLST scheme (*emrA, emrE, sugE, norM, clpA, clpP, stmPr1, htpX, tetACG, smoR, smeU2, sul1* and *sul2*). Multiple correspondence analysis (MCA) was performed on nine genes (*aac, aph, blaL1, blaL2, katA, pilU, qacE, smoR, sul1*) that were present or absent in at least 10 isolates as these were most likely to explain data set variance. The analysis was conducted using the factoextra and FactoMineR R packages[65].

**Statistical analysis and data management.** All statistical analyses and data management were performed in R version 3.4.3[66] using mainly packages included in the tidyverse[67], reshape2[68] and rcompanion (https://cran.r-project.org/web/packages/rcompanion/index.html). The map was created with the tmap package[69].

WgMLST data were analysed in BioNumerics v7.6.3 using the WGS and MLST plugins. For correlation testing between variables, Spearman's rank test was employed. The test of equal or given proportions or Fisher's exact test for sample sizes smaller than 5 was used to test for proportions. Association analysis was performed at the unitig level using the linear mixed model and whole-genome elastic net models as implemented in pyseer[70]. Unitigs were counted with unitig-counter[71], and a genetic similarity matrix was calculated from the core phylogeny. All isolates with unknown or anthropogenic isolation source were excluded from the analysis. Any unitig with a minor allele frequency of <1% was not tested. Control for population structure was verified with QQ plots using the packae qqman[72] (Supplementary Fig. 8). We additionally ran a whole-genome elastic net model where population structure was ensured by including sequence reweighting and supplying the BAPS groups as clonal complex designations.

**Reporting summary**. Further information on research design is available in the Nature Research Reporting Summary linked to this article.

## Data availability

The sequencing data generated and analysed during the current study are available in the European Nucleotide Archive repository under the accession numbers PRJEB32355, PRJEB32585 and PRJNA543082 (Supplementary Table 2). The Bacterial Antimicrobial Resistance Reference Gene Database is available under accession PRJNA313047. All data generated or analysed during this study are included in this published article and its supplementary information. The source data underlying Figs. 1a–c, 3, 4a, Supplementary Figs. 3b, 6 are provided as a Source Data File, along with supplemental Supplementary Data 3, 5. Supplementary Data 3 contains the sample metadata used to create Figs. 1d, 2, 5, and the tree annotation in Fig. 4a. Supplementary Data 5 comprises the allelic typing information for all samples and was used to create Figs. 4b, 5, and S7. The Supplementary Figs. 1 and 2 are based on data provided in Supplementary Data 4 and Supplementary Data 2, respectively. The wgMLST scheme is available from https://figshare.com/articles/Smaltophilia_wgMLST_all-alleles_fasta_gz/10005047.

## Code availability

Custom python scripts used in the phylogenetic analyses are available from https://github.com/conmeehan/pathophy. Custom R scripts used to calculate the statistical tests and perform the MCA analysis in Fig. 3b–d and Supplementary Fig. 5 are available from https://github.com/ngs-fzb/SMaltophilia_phylo and can be replicated using the source data files provided.

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

## Acknowledgements

We thank V. Mohr, F. Boysen and T. Ubben for technical assistance during next-generation sequencings. Parts of the work have been funded by grants from German Center for Infection Research, Federal Ministry of Education and Research, Germany, from Deutsche Forschungsgemeinschaft (DFG, German Research Foundation) under Germany's Excellence Strategy–EXC 22167-390884018, the Cluster Precision Medicine in Chronic Inflammation, and grants from the Leibniz Science Campus Evo-LUNG; I.A. and W.R.S. received funds from the University of Hamburg; C.F.L. and J.W.A.R. were funded through the European Commission Horizon 2020 Framework Marie Skłodowska-Curie Actions (Grant agreement number: 713660 - PRONKJE-WAIL - H2020-MSCA-COFUND-2015); C.J.M. was funded by the Department of Economy, Science and Innovation of the Flemish Government and the Belgian Science Policy (Belspo); K.Z. was funded by the National Natural Science Foundation of China (grant number 81702045), and 13th Five-Year National Major Science and Technology Projects of China (grant number 2018ZX10712001); O.C.S., D.Y., I.G. and X.D. were funded by the Spanish Ministry for Science, Innovation and Universities (grant reference BIO2015-66674-R); N.A.R. was funded by the SingHealth DUKE-NUS Pathology Academic Clinical Programme Clinical Innovation Support Grant (09/FY2017/P1/06-A20); U.N. was funded by the EU Horizon 2020 programme, grant agreement number 643476.

## Author contributions

M.I.G., J.R., J.S., S.N. and T.A.K. conceived the study. M.I.G., I.B., M.G., O.C.S., J.S. and T.A.K. curated the data. M.I.G., C.J.M., I.B., M.D., A.G., M.S., I.C.S., C.U., C.F.L., D.Y., M.R.F., O.C.S. and T.A.K. performed the formal analysis. J.R., J.S. and S.N. acquired funding. S.K., N.A.R., M.K., T.S.v.d.W., W.R.S., K.D.B., K.Z., T.S., I.A., W.S., K.Z., U.E.S., J.W.A.R., U.N., D.Y., I.G., X.D., J.R., U.M., M.R.F. and J.S. provided resources for this study. M.I.G., I.B., A.G. and C.F.L. visualised the results. M.I.G., J.S., S.N. and T.A.K. wrote the initial draft. All authors critically reviewed and modified the paper.

## Competing interests

The authors declare no competing interests.
