## [Peer Review File · Nature Communications]

Reviewers' Comments:

Reviewer #1:

Remarks to the Author:

Matthias I Gröschel and co-workers carried out a mega-scale genomic study with an aim to understand global phylogeny of *Stenotrophomonas maltophilia*, which is a WHO listed multidrug-resistant nosocomial pathogen. The authors have generated WGS data of 1,071 from this study and included WGS data of 234 isolates available from public repositories. Most of the strains were from Germany (932), USA (92), Australia (56) Switzerland (49) and Spain (42 strains). For 126 strains, no information was available on isolations source and hence was excluded. However, 117 strains were from the environmental origin, anthropogenic (52), Human invasive (n= 133), human non-invasive (n= 353) and human respiratory included 524 strains. Using 1274 core loci, they report the presence of 23 distinct monophyletic lineages, of which 17 were found to be novel lineages. Four lineages were found to be distantly related to complex or branch of remaining lineages. They analyzed Average Nucleotide identity of isolates within and between lineages to prove the presence of 23 lineages. One of the lineage (Sm6) was found to contain most of the strains (n=413) and it was most frequently found to be human-invasive and globally distributed with prominent markers of resistance and virulence. The authors also found an association of few lineages with environmental stains, anthropogenic, human respiratory track, human invasive and human non-invasive. The authors further compiled complete genome information of 15 major phylogenetic lineages and their analysis revealed considerable variation in structure and size between lineages and even within members of a particular lineage. Interestingly they find the absence of plasmid in all these complete genomes. Overall the large scale genomic study has revealed a major and globally lineage (Sm6) consisting of majorly human-associated strains that found in potential human-to-human transmission clusters with predicted high resistance to antibiotics and disinfectants. The findings stress the need for much larger scale studies in this direction to correlate genotype to patient outcomes to identify genomic groups that might be associated with higher virulence in a WHO listed MDR pathogen.

Below are major comments that need to be addressed before taking any decision on the manuscript
1) >80% of the collection is from just four countries of Europe and in fact more than 70% of from Germany. Title as a global study is misleading and hence potentially inconclusive.

2) Since sample collection is highly biased towards clinical isolates, genomic features findings related to human and environmental strains may be incorrect.

3) The authors have included WGS data from public repositories only up to April 2018. The authors have missed all the WGS data after April 2018 including one major and large scale taxonogenomic study of 29 clinical isolates from India

(<https://www.microbiologyresearch.org/content/journal/mgen/10.1099/mgen.0.000207>).

4) The author has used strain K279a as indicator strain of *S. maltophilia* lineage. However, from taxonomic point of view the reference or type strain of *S. maltophilia* is ATCC 13637(T) and its complete genome is available. Hence in this regard, use of K279a as reference strain is incorrect, particularly when the manuscript is pitched from taxonomic angle. Please see the below information from LPSN website (<http://www.bacterio.net/stenotrophomonas.html>)

(*Stenotrophomonas maltophilia* (Hugh 1981) Palleroni and Bradbury 1993, comb. nov. (Type species of the genus.)

Type strain: (see also Global Catalogue of Microorganisms) Stanier 67 = Hugh 810-2 = RH 1168 = AS 1.1788 = ATCC 13637 = CCUG 5866 = CFBP 3035 = CCM 1640 = BCRC (formerly CCRC) 10737 = CIP 60.77 = DSM 50170 = IAM 12423 = ICMP 17033 = IFO (now NBRC) 14161 = IMET 10402 = JCM 1975 = LMG 958 = NCAIM B.01119 = NCCB 68018 = NCIMB 9203 = NCPPB 1974 = NCTC 10257 = NRC 729 = NRRL B-2756 = VKM B-591.

Sequence accession no. (16S rRNA gene) for the type strain: AB294553.

Basonym: × *Pseudomonas maltophilia* (ex Hugh and Ryschenkov 1961) Hugh 1981.

Other synonyms: \times *Xanthomonas maltophilia* (Hugh 1981) Swings et al. 1983, "*Pseudomonas maltophilia*" Hugh and Ryschenkow 1961.

Etymology: N.L. n. *maltum*, malt; Gr. n. *philia*, friendship; N.L. n. *maltophilia*, intended to mean friend of malt.

Valid publication: PALLERONI (N.J.) and BRADBURY (J.F.): *Stenotrophomonas*, a new bacterial genus for *Xanthomonas maltophilia* (Hugh 1980) Swings et al. 1983. *Int. J. Syst. Bacteriol.*, 1993, 43, 606-609.)

5) In this context formal proposal of *Stenotrophomonas maltophilia* complex is invalid. Unless each or many of these novel lineages or the potential novel species are formally published with formal nomenclature along with biochemical data and if possible associated data like FAME analysis and morphological analysis. Further, the authors have not included the WGS data of all available type strains of the genus *Stenotrophomonas* to identify potential novel and known species lineages from their study.

Also in the taxonogenomics study of a large number of clinical studies using all available reference strain or type strains of genus *Stenotrophomonas*, the terminologically is repeated referred in the said publication (<https://www.microbiologyresearch.org/content/journal/mgen/10.1099/mgen.0.000207>).

6) In above-mentioned work using type strain coupled taxonomy studies on clinical isolates from India suggests the presence of only two valid species (*S. maltophilia* and *S. pavanii*) and four misclassified species (*P. hibiscicola*, *P. geniculata*, *P. betele* and *S. africana*) and belonging to the Smc. Further this work also report that five novel genomospecies/lineages among clinical isolates. It is not quite surprising that in present work, authors have found 23 monophyletic lineages as previous studies already described a diverse nature of *S. maltophilia* clinical isolates and inclusion of more genomes into dataset from the different geographical region will add a number of new lineages/ genomospecies.

7) Previous studies already reported that core *S. maltophilia* lineage Sm6 is dominant among clinical isolates, which is again reflecting into global phylogeny.

8) Information on completeness and contamination of the genomes through software such as CheckM is missing. Without this information, the analysis will be incomplete.

9) Phylogenetic analysis and previous studies revealed the *S. maltophilia* clinical strains are highly diverse and the species consist of hidden species. Essentially authors studies suggest 23 species lineages in this organism. Hence SNP analysis is perfectly fine within species but may not be correct across diverse species. Hence the authors need to confirm the presence of 23 groups using protein-based trees rather than nucleotide-based trees.

10) The authors suggest some lineages as ancestral and one lineage as most distant. Such inference is not valid without outgroup like *Xanthomonas*, etc.

Reviewer #2:

Remarks to the Author:

The manuscript, "The global phylogenetic landscape and nosocomial spread of the Multidrug-resistant opportunist *Stenotrophomonas maltophilia*," describes a population genomic analysis of 1305 *S. maltophilia* whole-genome sequences representing isolates from 22 countries across six continents. The authors define a novel wgMLST scheme and use it to identify core genes for maximum-likelihood phylogenetic analysis and Bayesian Analysis of Population Structure. A total of 23 distinct and well supported phylogenetic lineages are defined including several novel lineages. Many of these are broadly geographically distributed. Lineage Sm6 was the most common among human-infection isolates globally while lineages Sgn1-4 (outside of the *S. maltophilia* complex) were associated with environmental isolates and less likely to carry key virulence and drug resistance associated genes.

The data and analyses provide insights into the population structure of *S. maltophilia*, a multi-drug resistant opportunistic human pathogen. However, I note that it has been shown previously that

human-associated *S. maltophilia* form distinct phylogenetic groups from environmental isolates e.g. Steinmann et al Front Microbiol 2018; Ochoa-Sanchez et al Front Microbiol 2017, although these previous analyses comprised considerably fewer isolates and did not capture the full extent of genomic diversity shown here. Additionally, Sm6 has been shown to account for the greatest burden of human infections- as the authors note.

The novel wgMLST scheme and lineage nomenclature developed herein will provide a highly valuable resource for future exploration of the epidemiology and transmission of this organism. However, its impact and utility would be greatly enhanced if the full wgMLST scheme information were made publicly available in an open source repository rather than provided solely as a Bionumerics plugin (to my knowledge the Bionumerics software requires a paid licence).

I enjoyed reading the manuscript, I appreciate the amount of work and value of a genome collection of this size. However, I don't feel as though the data have yet been leveraged to their full potential in order to generate substantially novel findings over the previously published comparative genomic analyses (e.g. see comments below regarding gene content analyses).

I have a few major comments/suggestions about the data and analyses and a number of minor comments.

MAJOR COMMENTS:

Relating to data availability and reproducibility of the wgMLST method:

As well as making the wgMLST scheme publicly available it would be helpful to include the following information in the supplementary data;

- 1) the full set of allelic typing information for all genomes included in the analysis
- 2) notation in the supplementary table to show the loci that were used for the core genome alignment for phylogenetic analyses, and to note the 7-gene MLST loc
- 3) information about resistance and virulence gene detection in the Supplementary table detailing all of the genomes included in the analysis.
- 4) more detailed collection date and location information for the newly sequenced isolates – this is required to understand and reproduce the findings about closely related groups of strains collected over short time-periods in the same location i.e. potential transmission clusters.

Relating to section titled: "Genomic features of human-associated and environmental *S. maltophilia* lineages" and fig 2c. I don't think this analysis is particularly helpful. The number of unique genes to a given type of strains is not particularly informative on its own and will be under strong influence of the underlying population structure. If the authors would like to identify loci associated with human disease or particular disease types they should use a statistical association analysis approach that includes correction for population structure e.g as implemented in BUGWAS or dbGWAS. Such an analysis can only be achieved with a large dataset such as this one and would greatly enhance the novelty of the manuscript.

Related to this- The extent of the pan-genome diversity is intriguing given the lack of plasmids in the completed genomes- Are the accessory genes very structured in the population? I.e. does the gene content diversity simply reflect the accumulation of losses and gains since divergence of the lineages (could explain the high numbers of unique genes per lineage?) or is there evidence for HGT between lineages? How much gene content variation is there within lineages compared to that between them? This sort of information would also enhance the novelty of the work, but is not essential to support the authors conclusions.

MINOR COMMENTS:

Introduction:

1. Liens 81-84: "S. maltophilia is an important cause of hospital-acquired drug-resistant infections with a significant attributable mortality rate in immunocompromised patients of up to 37.5% 7–9." Please ensure the correct primary references are used here and throughout- Steinmann et al 2018 does not report mortality rates, however it does reference Falagas et al 2009; Attributable mortality of *Stenotrophomonas maltophilia* infections: a systematic review of the literature.

Results:

2. Liens 130-132: Across the 1305 strains, most loci, 13,002 of 17,603, were assigned fewer than 50 different alleles, consistent with a large accessory genome (Fig. S2)." – I don't quite understand the logic for this statement- how is allelic variation related to the scale of the accessory genome? Or perhaps you mean that most loci were detected in 50 or fewer genomes? Please clarify in the text.
3. Lines 139-141: "We recovered between 380 loci in *S. dokdonensis* to a maximum of 1,677 in *S. rhizophila*, with species obtaining." Is the last part of this sentence missing? Please check.
4. Lines 143-145: "Interestingly, the 16S rRNA gene sequence of JCM9942 matched to *S. acidaminiphila*, and the JCM9942 16S rRNA sequence is only 97.3% identical with that of *S. maltophilia* K279a" Is the inference here that JCM9942 is incorrectly labelled? What is the ANI compared to K279a?
5. Lines 192-195: "The more distantly related lineages Sgn1 (85%), Sgn2 (83%), Sgn3 (74%), and also Sm11 (34%) contained significantly more environmental strains ($p < .001$, test of equal or given proportions or Fisher's exact test for $n < 5$), whereas strains of lineages Sm4a and Sm6 (2% and 5%, $p < .001$) were minority environmental (Fig. 2B and 2D, Fig. 4, Table S3)" Please clarify the comparison here- i.e. is each lineages compared to each other lineage or to the rest of the genomes as a whole? Please also indicate if and which multiple-testing correction has been applied to the p-values that are given.
6. Line 212: "we examined genes unique to strains isolates from different sources" Typo here- strains or isolates?
7. Line 255-256: "some of which are unequally distributed over the lineages." Please add numbers to support this statement to the text above and where appropriate statistical comparisons. E.g. for PilU the authors state that it was mostly found in Sm9 and Sm11 but what proportion of the positive isolates were accounted for by these lineages? Are there any other key virulence/resistance genes for which the proportions of positive genomes differs substantially between lineages? What are the proportions? Etc.
8. Lines 257-277: In discussion of the MCA results please give the value for the amount of variance captured in the analysis in the text and specify the values for each of dimensions 1 and 2, which are discussed in more detail.
9. Lines 274-277: "A more detailed analysis of the observed lineage-specific variation reveals that the more distantly related lineages Sgn1-4 are characterized by the lack of *smoR*, *katA*, and *blaL2* virulence and resistance determinants, whereas the human-associated lineages Sm6, Sm9, and Sm11 are strongly associated with the presence of *blaL2*, *aph*, *blaL1*, *smoR*, and *katA* (Fig. 3C)." Please clarify if this statement is based on a quantitative assessment and comparison or visual inspection of the MCA plots?

Discussion:

10. Line 346: "likely from an exclusively environmental lifestyle towards human colonization and infection." Please expand a little on the rationale here and/or provide references. Could it be possible that *S. maltophilia* has evolved primarily as a host-associated organism and then moved into environmental niches?
11. Lines 366-370: "In contrast, the most successful human-associated lineage Sm6 was linked to the presence of β -lactamases (*BlaL1* and *BlaL2*) and aminoglycoside resistance-conferring enzymes (*Aph*)

as well as KatA, involved in resistance to disinfectants, pointing towards adaptation to healthcare settings and survival on and in patients." I think this statement is a little misleading- by highlighting Sm6 in this way a reader may think that Sm6 is special in harbouring this list of genes, but most are present in a large proportion of the Sm complex. However, perhaps it is the combination of all of these that is particularly important in Sm6? Is it the only lineage with high prevalence of all of the named genes?

12. Lines 373-374: "This notion is also supported by our finding that we did not detect any d100 clusters, or circulating variants, in the primarily environmental-associated lineages." The authors should mention here the potential role of differences in sample size and the lack of systematic samples for environmental isolates. Would we reasonably expect to find any groups of closely related clusters with the number of samples available? Particularly given that the numbers from any single location and time-frame will be low.

13. Line 398: "lineage Sm6 strains potentially best adapted to colonize or infection humans." – Typo- Colonise or infect humans

Methods:

14. Please specify how many SNPs were used for the BAPS and phylogenetic analyses i.e. the alignment length.

15. Please expand the rationale for including urine isolates among the invasive infections rather than non-invasive? They are most commonly considered non-invasive. Could this have influenced the results and conclusions in any way?

16. Please give more specific details about the methods for obtaining consensus allele calls e.g. coverage and identify cut-offs or k-mer depth thresholds. Please also give more details on the statement: "Biological replicates (sequencing data obtained from different fresh cultures of *S. maltophilia* strain ATCC 1363752) differed at maximum 5 consensus allele calls." This seems like a high number of differences for sequence data from recent subcultures of a single reference strain. What was causing the differences here? i.e. does this reflect a lack of allele calls in some genomes compared to others e.g. due to differences in sequencing depth? Or does this reflect allelic mismatches? The latter is more concerning as it calls into question the reproducibility of the results and is relevant to understanding the cluster analyses.

17. Please state the number of bootstrap replicates used in the phylogenetic analysis.

18. Please indicate how many SNPs were masked from the alignment after ClonalFrame analysis.

Other:

19. Figure 1a: I find the layout of the tree here a little confusing. It looks like the branches leading to Sgn4 and its sister clade have been truncated and moved to fit into the figure (?) but this is not obvious at first glance and not highlighted in the figure legend. I suggest rethinking the layout. Perhaps it would work to show a zoomed out view of the full tree (with all branches to scale and in true position) alongside a zoomed in view of just *S. maltophilia* sensu lato?

20. Fig 1d and 2a: Please consider reordering the lineage labels so that they are shown in numerical order (and coloured correspondingly).

Reviewers' comments:

Reviewer #1 (Remarks to the Author):

Matthias I Gröschel and co-workers carried out a mega-scale genomic study with an aim to understand global phylogeny of *Stenotrophomonas maltophilia*, which is a WHO listed multidrug-resistant nosocomial pathogen. The authors have generated WGS data of 1,071 from this study and included WGS data of 234 isolates available from public repositories. Most of the strains were from Germany (932), USA (92), Australia (56) Switzerland (49) and Spain (42 strains). For 126 strains, no information was available on isolations source and hence was excluded. However, 117 strains were from the environmental origin, anthropogenic (52), Human invasive (n= 133), human non-invasive (n= 353) and human respiratory included 524 strains. Using 1274 core loci, they report the presence of 23 distinct monophyletic lineages, of which 17 were found to be novel lineages. Four lineages were found to be distantly related to complex or branch of remaining lineages. They analyzed Average Nucleotide identity of isolates within and between lineages to prove the presence of 23 lineages. One of the lineage (Sm6) was found to contain most of the strains (n=413) and it was most frequently found to be human-invasive and globally distributed with prominent markers of resistance and virulence. The authors also found an association of few lineages with environmental stains, anthropogenic, human respiratory track, human invasive and human non-invasive. The authors further compiled complete genome information of 15 major phylogenetic lineages and their analysis revealed considerable variation in structure and size between lineages and even within members of a particular lineage. Interestingly they find the absence of plasmid in all these complete genomes. Overall the large scale genomic study has revealed a major and globally lineage (Sm6) consisting of majorly human-associated strains that found in potential human-to-human transmission clusters with predicted high resistance to antibiotics and disinfectants. The findings stress the need for much larger scale studies in this direction to correlate genotype to patient outcomes to identify genomic groups that might be associated with higher virulence in a WHO listed MDR pathogen.

We thank reviewer #1 for the kind words emphasizing the scope of our study, for critically reading our manuscript, and all the helpful comments that we address below.

Below are major comments that need to be addressed before taking any decision on the manuscript

1) >80% of the collection is from just four countries of Europe and in fact more than 70% of from Germany. Title as a global study is misleading and hence potentially inconclusive.

We agree with the reviewer and have removed 'global' from the title.

2) Since sample collection is highly biased towards clinical isolates, genomic features findings related to human and environmental strains may be incorrect.

This is indeed an important point and one of the major limitations of our study, as indicated in the corresponding paragraph in the discussion. Apart from the potential bias by only collecting clinical isolates, we also discuss that there might be a bias among these clinical isolates since no routine (molecular) surveillance of *S. maltophilia* is in place in the countries where the strains have been collected. For that reason, we do not wish to propose definitive gene demarcations between isolation sources as we try to show that e.g. resistance genes are found at higher proportions in lineage Sm6. In addition and in agreement with the reviewer's comment, we have removed the respective sub-analysis of genomic features from human-associated vs. environmental lineages.

3) The authors have included WGS data from public repositories only up to April 2018. The authors have missed all the WGS data after April 2018 including one major and large scale taxonogenomic study of 29 clinical isolates from India (<https://www.microbiologyresearch.org/content/journal/mgen/10.1099/mgen.0.000207>).

We appreciate the genus level study by Patil P *et al.* which we also cited in our manuscript. As we had to choose a date up to which to include new public sequences for our downstream analyses, we were unable to include them in our study. In light of the reviewer's comment, and to enable correlating our results with the findings of Patil P *et al.*, we have run the sequence data of this study through the wgMLST pipeline and have added Source Data File 1 that contains all allele calls for all isolates, including these new sequences. We also included these new sequences in a new protein based tree (Fig. S5) to show which lineage they are part of.

4) The author has used strain K279a as indicator strain of *S. maltophilia* lineage. However, from taxonomic point of view the reference or type strain of *S. maltophilia* is ATCC 13637(T) and its complete genome is available. Hence in this regard, use of K279a as reference strain is incorrect, particularly when the manuscript is pitched from taxonomic angle. Please see the below information from LPSN website (<http://www.bacterio.net/stenotrophomonas.html>)
(*Stenotrophomonas maltophilia* (Hugh 1981) Palleroni and Bradbury 1993, comb. nov. (Type species of the genus.) Type strain: (see also Global Catalogue of Microorganisms) Stanier 67 = Hugh 810-2 = RH 1168 = AS 1.1788 = ATCC 13637= CCUG 5866 = CFBP 3035 = CCM 1640 = BCRC (formerly CCRC) 10737 = CIP 60.77 = DSM 50170 = IAM 12423 = ICMP 17033 = IFO (now NBRC) 14161 = IMET 10402 = JCM 1975 = LMG 958 = NCAIM B.01119 = NCCB 68018 = NCIMB 9203 = NCPPB 1974 = NCTC 10257 = NRC 729 = NRRL B-2756 = VKM B-591.
Sequence accession no. (16S rRNA gene) for the type strain: AB294553.
Basonym: α *Pseudomonas maltophilia* (ex Hugh and Ryschenkov 1961) Hugh 1981.
Other synonyms: α *Xanthomonas maltophilia* (Hugh 1981) Swings *et al.* 1983, "*Pseudomonas maltophilia*" Hugh and Ryschenkov 1961.
Etymology: N.L. n. *maltum*, malt; Gr. n. *philia*, friendship; N.L. n. *maltophilia*, intended to mean friend of malt.
Valid publication: PALLERONI (N.J.) and BRADBURY (J.F.): *Stenotrophomonas*, a new bacterial genus for *Xanthomonas maltophilia* (Hugh 1980) Swings *et al.* 1983. *Int. J. Syst. Bacteriol.*, 1993, 43, 606-609.)

We thank the reviewer for pointing this out. We have indeed included the type strain of *S. maltophilia* (ATCC 13637) in all our analysis using its GenBank assembly accession (GCA_000742995.1), and we used the 16S rRNA sequence of ATCC 13637 to ensure that all our sequences were indeed *S. maltophilia*.

As we are interested in the prevalence of *S. maltophilia* in healthcare settings and the molecular population structure, we used K279a locus tags for the wgMLST scheme, as they are widely used among wet lab researchers, and most biological insights were obtained using K279a as a model strain.

In accordance with the excellent point made by the reviewer, we have removed all instances in the manuscript where K279a is suggested as type or species reference strain, and made clear that we used it as a model of a hospital associated strain.

5) In this context formal proposal of *Stenotrophomonas maltophilia* complex is invalid. Unless each or many of these novel lineages or the potential novel species are formally published with formal

nomenclature along with biochemical data and if possible associated data like FAME analysis and morphological analysis. Further, the authors have not included the WGS data of all available type strains of the genus *Stenotrophomonas* to identify potential novel and known species lineages from their study. Also in the taxonogenomics study of a large number of clinical studies using all available reference strain or type strains of genus *Stenotrophomonas*, the terminologically is repeated referred in the said publication

(<https://www.microbiologyresearch.org/content/journal/mgen/10.1099/mgen.0.000207>).

In this project, we did not aim to propose the *S. maltophilia* complex. We show the *S. maltophilia* complex in figure 1 as it is currently used in the literature (i.e. Svensson-Statdler *et al.*, 2011, and Vineusa P. *et al*, 2018, Patil P. *et al*, 2018), and how this directly relates to our results. In contrast, we argue that the *S. maltophilia* complex *sensu lato* and *sensu stricto* terminology should not be used as even lineages outside the proposed complex (Sgn 1-4), and lineages outside the complex *sensu stricto* (i.e. lineage Sm6) can be human associated. As an important point, our study does not suggest any novel species for the genus.

6) In above-mentioned work using type strain coupled taxonomy studies on clinical isolates from India suggests the presence of only two valid species (*S. maltophilia* and *S. pavanii*) and four misclassified species (*P. hibiscicola*, *P. geniculata*, *P. betele* and *S. africana*) and belonging to the Smc. Further this work also report that five novel genomospecies/lineages among clinical isolates. It is not quite surprising that in present work, authors have found 23 monophyletic lineages as previous studies already described a diverse nature of *S. maltophilia* clinical isolates and inclusion of more genomes into dataset from the different geographical region will add a number of new lineages/ genomospecies.

We appreciate all previous work on the various lineages which we cite in the introduction when introducing the *S. maltophilia* complex and the many already discovered lineages:

“Previous work indicated the presence of at least 13 lineages or species-like lineages in the *S. maltophilia* complex, defined as *S. maltophilia* strains with 16S rRNA gene sequence similarities > 99.0%, with nine of these potentially human-associated^{8,16–19}”

We hoped to harmonize lineage terminology by using previously published group names (Vinueza *et al*, 2018, Ochoa Sanchez *et al* 2017), and by showing how the Patil P *et al* genomes fall into these lineages (Fig. S5).

In this project, our aim was not to declare new genomospecies or species-like groups. We instead refer to lineages, directly building on and extending the previous work mentioned by the reviewer. With this in mind, our main focus was to investigate the prevalence and molecular epidemiology of clinically associated *S. maltophilia* complex strains in the background of their population structure.

7) Previous studies already reported that core *S. maltophilia* lineage Sm6 is dominant among clinical isolates, which is again reflecting into global phylogeny.

We agree with reviewer 1 that our findings are in agreement with and extend previous reports. Thanks to these studies and by including the sequence data generated there, we were able to analyze the *S. maltophilia* phylogeny at unprecedented large scale. Our study extends prior knowledge that nearly all lineages are associated with human infection and colonization. By employing a new versatile tool, wgMLST, we were able to detect transmission events among genetically closely related strains. In line with this comment, we also believe that more lineages are likely to be discovered as more *S. maltophilia* strains are being sequenced in the future.

8) Information on completeness and contamination of the genomes through software such as CheckM is missing. Without this information, the analysis will be incomplete.

We thank reviewer 1 for this important and helpful suggestion which we incorporated in our analyses (this information has been added to suppl. data S4). We note that strains from one study using metagenomic sequencing (Parks D *et al*, Nat Microbiol, 2017) showed <90% completeness in CheckM analyses. However we ensured that these strains had passed all other quality filtering steps, including enough allele calls in the wgMLST analyses, to justify keeping them in our analyses.

9) Phylogenetic analysis and previous studies revealed the *S. maltophilia* clinical strains are highly diverse and the species consist of hidden species. Essentially authors studies suggest 23 species lineages in this organism. Hence SNP analysis is perfectly fine within species but may not be correct across diverse species. Hence the authors need to confirm the presence of 23 groups using protein-based trees rather than nucleotide-based trees.

In this project, we do not propose new species, in contrast, by using the term lineages we are trying not to get involved in taxonomy discussions, but rather provide a molecular epidemiology snapshot of *S. maltophilia*'s population structure. We recognize the need for a proper, genome-based taxonomy evaluation of the genus *S. maltophilia* which we plan to undertake and will reach out to others in the field to conduct such analyses collaboratively in the near future.

We have not used SNP analyses in this study. Our results rely on core gene sequence alignments. We have amended supplementary data S2 with the core loci that were used for the alignment. Although protein trees have been used for within-genus phylogenetics, the close evolutionary distance between genomes means that core gene alignments are also sufficient. We have carefully constructed core gene alignments for the work here and are confident these are able to reliably reconstruct the relationships between lineages.

10) The authors suggest some lineages as ancestral and one lineage as most distant. Such inference is not valid without outgroup like *Xanthomonas*, etc.

We describe our findings on the phylogeny of the *S. maltophilia* complex based on previous literature in the introduction, where groups Sgn1 - Sgn4 were found to be more distantly placed in the phylogeny than the *S. maltophilia* complex (*sensu stricto* and *sensu lato*) (Vinuesa *et al.*, 2018, Ochoa Sanchez *et al.*, 2017).

In the *results* section, we replicate the finding that some lineages are more distantly placed to the other *S. maltophilia* complex strains, yet we do not imply that these are more ancestral.

“In concordance with these studies^{16,18}, we found a clear separation of the more distantly related lineages Sgn1-Sgn4 and a branch formed by lineages Sm1-Sm18 (previously termed *S. maltophilia sensu lato*), with the largest lineage Sm6 (also known as *S. maltophilia sensu stricto*) containing most strains (n = 413) including the strain K279a and the species type strain ATCC 13637. Contrary to previous analyses, Sgn4 is the lineage most distantly related to the rest of the strains¹⁶.”

We believe that a proper taxonomy follow up study collaboratively with other groups in this space is needed to clarify the taxonomic placement of the various groups / lineages / species found in previous work.

Reviewer #2 (Remarks to the Author):

The manuscript, "The global phylogenetic landscape and nosocomial spread of the Multidrug-resistant opportunist *Stenotrophomonas maltophilia*," describes a population genomic analysis of 1305 *S. maltophilia* whole-genome sequences representing isolates from 22 countries across six continents. The authors define a novel wgMLST scheme and use it to identify core genes for maximum-likelihood phylogenetic analysis and Bayesian Analysis of Population Structure. A total of 23 distinct and well supported phylogenetic lineages are defined including several novel lineages. Many of these are broadly geographically distributed. Lineage Sm6 was the most common among human-infection isolates globally while lineages Sgn1-4 (outside of the *S. maltophilia* complex) were associated with environmental isolates and less likely to carry key virulence and drug resistance associated genes.

The data and analyses provide insights into the population structure of *S. maltophilia*, a multi-drug resistant opportunistic human pathogen. However, I note that it has been shown previously that human-associated *S. maltophilia* form distinct phylogenetic groups from environmental isolates e.g. Steinmann et al Front Microbiol 2018; Ochoa-Sanchez et al Front Microbiol 2017, although these previous analyses comprised considerably fewer isolates and did not capture the full extent of genomic diversity shown here. Additionally, Sm6 has been shown to account for the greatest burden of human infections- as the authors note.

We thank reviewer #2 for critically examining our manuscript and all the helpful suggestions. Thanks to previous studies we were able to analyze the phylogeny with unprecedented large scale and diverse collection and employed a new versatile tool, wgMLST, enabling us to extend and substantiate prior knowledge.

The novel wgMLST scheme and lineage nomenclature developed herein will provide a highly valuable resource for future exploration of the epidemiology and transmission of this organism. However, its impact and utility would be greatly enhanced if the full wgMLST scheme information were made publicly available in an open source repository rather than provided solely as a Bionumerics plugin (to my knowledge the Bionumerics software requires a paid licence).

We fully agree with reviewer #2 and we provide the loci of the wgMLST scheme (https://figshare.com/articles/Smaltophilia_wgMLST_all-alleles_fasta_gz/10005047). Any wgMLST scheme will only be a valuable addition to the community if it can be accessed freely by everyone. We hope that our work provides the basis for MLST resources such as BIGSdb and chewBBACA to ensure that wgMLST using this scheme can be done reliably, and to ensure that the nomenclature of allele call numbers is synchronized across the platforms. Of course, we will be happy to support this further and in full.

I enjoyed reading the manuscript, I appreciate the amount of work and value of a genome collection of this size. However, I don't feel as though the data have yet been leveraged to their full potential in order to generate substantially novel findings over the previously published comparative genomic analyses (e.g. see comments below regarding gene content analyses).

I have a few major comments/suggestions about the data and analyses and a number of minor comments.

MAJOR COMMENTS:

Relating to data availability and reproducibility of the wgMLST method:

As well as making the wgMLST scheme publicly available it would be helpful to include the following information in the supplementary data;

1) the full set of allelic typing information for all genomes included in the analysis

We provide all allelic typing information as supplemental data S5, including data on the recently published *S. maltophilia* strains by Patil P *et al* (2018).

2) notation in the supplementary table to show the loci that were used for the core genome alignment for phylogenetic analyses, and to note the 7-gene MLST loc

We amended supplemental data S2 with the following columns: locus tag (containing the orthologous gene tags from K279a or other annotations), 99 percent core genome (representing all loci used for the core gene alignment) and 7 gene MLST (detailing the 7 genes that are part of the classical MLST scheme).

Please note that this information is also provided in table S1 “Details of the 7 classical Multilocus Sequence Typing loci”.

3) information about resistance and virulence gene detection in the Supplementary table detailing all of the genomes included in the analysis.

We provide a new Source supplemental data S5 that lists presence (1) or absence (0) of the resistance genes used in this study. We added details to the methods on which genes were obtained through a literature search and subsequently pulling the presence/absence data from the wgMLST results in addition to the genes identified using the program *abricate* (that screens several public resistance and virulence gene databases).

4) more detailed collection date and location information for the newly sequenced isolates – this is required to understand and reproduce the findings about closely related groups of strains collected over short time-periods in the same location i.e. potential transmission clusters.

We added this information in two columns ('city' and 'collection day') in supplemental data S3 that were used for this analysis.

Relating to section titled: “Genomic features of human-associated and environmental *S. maltophilia* lineages” and fig 2c. I don't think this analysis is particularly helpful. The number of unique genes to a given type of strains is not particularly informative on its own and will be under strong influence of the underlying population structure. If the authors would like to identify loci associated with human disease or particular disease types they should use a statistical association analysis approach that includes correction for population structure e.g as implemented in BUGWAS or dbGWAS. Such an analysis can only be achieved with a large dataset such as this one and would greatly enhance the novelty of the manuscript.

We agree that this analysis had a rather exploratory character and does not provide a lot of insights as it stands now. In line with a corresponding comment from reviewer 1, we therefore decided to drop this analysis from the manuscript.

We have invested the past months to obtain information on genes (presence or absence) as well as k-mers and unitigs associated with human infection/colonization using a linear mixed model (LMM) as well as a whole genome elastic net model as implemented in pyseer. The

inflated nature of the resulting QQ plots led us to conclude that the population structure and niche-specificity of the lineages are too strong confounders in this analysis. We present these results in Fig S7 and in the *results* section:

“We attempted genome wide association (GWAS) to investigate the genetic correlates of human niche specificity using an elastic net whole genome model that has recently been shown to outperform univariate approaches in controlling for population structure²⁷. Using this approach there was still considerable confounding due to residual population structure, likely related to the strong niche specificity of some of the lineages described above.”

We also discuss these findings in the *discussion*:

“Beyond associating some of the lineages with either environment or human, we were not able to identify the specific genetic mechanisms that underlie this association due to the extent of stratification by population structure. *S. maltophilia* is believed to be a much less virulent pathogen relative to other nosocomials such as *P. aeruginosa* or *S. aureus*³⁸. The establishment of human infection or colonization with *S. maltophilia* is likely also strongly driven by the host immune status while the role of pathogen genetic background or specific virulence mechanisms is still to be determined³⁹. Collecting data on the host immune status or other predisposing factors will enable research in this area in the future.”

This is - in addition to the comment made by the reviewer - to also recognize that the concept of a ‘human-associated pathogen’ in the context of *S. maltophilia* is controversial. *S. maltophilia* is considered a lower-grade pathogen compared to high-grade nosocomials such as *P. aeruginosa* and *S. aureus*. Although isolation of low-grade pathogenic microorganisms from human samples is relatively rare, the threat with *S. maltophilia* is that once colonization is established they are difficult to eradicate despite not having specific virulence mechanisms in humans. The difficulty in eradicating these colonizing opportunists could stem from their ability to form biofilms or from their inherent resistance to a wide range of antibiotics (or both). Here, the presence/absence analysis of virulence and resistance determinants is already included in our work where we concluded that lineages harbouring mostly environmental strains tended to harbour less resistance and virulence genes than lineages being comprised mostly out of human-associated strains.

Related to this- The extent of the pan-genome diversity is intriguing given the lack of plasmids in the completed genomes- Are the accessory genes very structured in the population? I.e. does the gene content diversity simply reflect the accumulation of losses and gains since divergence of the lineages (could explain the high numbers of unique genes per lineage?) or is there evidence for HGT between lineages? How much gene content variation is there within lineages compared to that between them? This sort of information would also enhance the novelty of the work, but is not essential to support the authors conclusions.

We agree with the reviewer that these are intriguing research questions. We used prokka and roary to add a supplemental figure (Fig. S3b) showing a gene presence / absence matrix to give an impression of the structure of the pan genome. From our results, we see no clear gene structure in the population. Given the surprising amount of variability we found in completed genomes, we feel that detailed work in this direction would require a fully dedicated study and more data, especially fully finished genomes. We would be delighted if our work could in fact provide the basis for this kind of analysis.

MINOR COMMENTS:

Introduction:

1. Liens 81-84: “*S. maltophilia* is an important cause of hospital-acquired drug-resistant infections with a significant attributable mortality rate in immunocompromised patients of up to 37.5% 7–9.” Please ensure the correct primary references are used here and throughout- Steinmann et al 2018 does not report mortality rates, however it does reference Falagas et al 2009; Attributable mortality of *Stenotrophomonas maltophilia* infections: a systematic review of the literature.

We modified the references and now only cite the primary reference at the end of the sentence (Falagas *et al.*, 2009).

Results:

2. Liens 130-132: Across the 1305 strains, most loci, 13,002 of 17,603, were assigned fewer than 50 different alleles, consistent with a large accessory genome (Fig. S2).” – I don’t quite understand the logic for this statement- how is allelic variation related to the scale of the accessory genome? Or perhaps you mean that most loci were detected in 50 or fewer genomes? Please clarify in the text.

We agree with this remark and have removed the second part of the sentence. Indeed, the number of different alleles does not indicate the size of the pan genome, i.e. in organisms with a limited accessory/pan genome one could still find that most loci have only few different alleles, indicating low genetic diversity. Our reasoning of the initial statement was that low numbers of different alleles could be explained by many accessory genes that occur only rarely across the isolate collection.

3. Lines 139-141: “We recovered between 380 loci in *S. dokdonensis* to a maximum of 1,677 in *S. rhizophila*, with species obtaining.” Is the last part of this sentence missing? Please check.

We removed these two words that seem to be left over from previous editing. We thank reviewer #2 for spotting this.

4. Lines 143-145: “Interestingly, the 16S rRNA gene sequence of JCM9942 matched to *S. acidaminiphila*, and the JCM9942 16S rRNA sequence is only 97.3% identical with that of *S. maltophilia* K279a” Is the inference here that JCM9942 is incorrectly labelled? What is the ANI compared to K279a?

This is indeed an interesting question. According to our results, we can distinguish genomes belonging to the *S. maltophilia* complex and other *Stenotrophomonas* species by using a threshold of 2000 allele calls (Fig. S1h), also supporting the conclusion that JCM9942 is incorrectly labeled. The ANI between JCM9942 and the type strain MTCC 434 is 82.981. However, in this study we refrain from proposing any taxonomic reclassifications as we think this should be done in a formal taxonomic study.

5. Lines 192-195: “The more distantly related lineages Sgn1 (85%), Sgn2 (83%), Sgn3 (74%), and also Sm11 (34%) contained significantly more environmental strains ($p < .001$, test of equal or given proportions or Fisher’s exact test for $n < 5$), whereas strains of lineages Sm4a and Sm6 (2% and 5%, $p < .001$) were minority environmental (Fig. 2B and 2D, Fig. 4, Table S3)” Please clarify the comparison here- i.e. is each lineages compared to each other lineage or to the rest of the genomes as a whole? Please also indicate if and which multiple-testing correction has been applied to the p-values that are given.

We added in the main text the multiple-testing correction used (Benjamini Hochberg) which was also stated in the methods section and in the supplement table S3. For the comparisons, we created a 2x2 contingency table for each lineage / isolation source combination, based on the supplement table S3, comparing all strains from one lineage (i.e. Sgn1) and one source (i.e. environmental) to all other strains of this group from all other sources (excluding environmental in this example).

We expanded the first sentence in this paragraph to make the comparisons performed here clearer. We also noted that we had given the percentages of isolation sources per lineage based while including strains of unknown source. However, we had calculated the association lineage-source without these strains, as shown in suppl. table S3. We corrected the percentages in the main text accordingly.

6. Line 212: "we examined genes unique to strains isolates from different sources" Typo here- strains or isolates?

We deleted this paragraph as we agree with the reviewer that this analysis will be more valuable if linked to a statistical genetics (GWAS) analysis.

7. Line 255-256: "some of which are unequally distributed over the lineages." Please add numbers to support this statement to the text above and where appropriate statistical comparisons. E.g. for PilU the authors state that it was mostly found in Sm9 and Sm11 but what proportion of the positive isolates were accounted for by these lineages? Are there any other key virulence/resistance genes for which the proportions of positive genomes differs substantially between lineages? What are the proportions? Etc.

We added the proportions and test results (test of equal or given proportions) for the genes discussed in the results for the lineages that had the highest / lowest proportions compared to the entire dataset.

8. Lines 257-277: In discussion of the MCA results please give the value for the amount of variance captured in the analysis in the text and specify the values for each of dimensions 1 and 2, which are discussed in more detail.

We added this information in the respective results paragraph.

9. Lines 274-277: "A more detailed analysis of the observed lineage-specific variation reveals that the more distantly related lineages Sgn1-4 are characterized by the lack of smoR, katA, and blaL2 virulence and resistance determinants, whereas the human-associated lineages Sm6, Sm9, and Sm11 are strongly associated with the presence of blaL2, aph, blaL1, smoR, and katA (Fig. 3C)." Please clarify if this statement is based on a quantitative assessment and comparison or visual inspection of the MCA plots?

We added in the respective sentence that this conclusion is based on the quantitative output from the MCA analysis, i.e. from a quantitative assessment.

Discussion:

10. Line 346: "likely from an exclusively environmental lifestyle towards human colonization and infection." Please expand a little on the rationale here and/or provide references. Could it be possible that *S. maltophilia* has evolved primarily as a host-associated organism and then moved into environmental niches?

We add two sentences to expand on the rationale and provide an additional reference that has investigated niche adaptation in *Legionella pneumophila*.

11. Lines 366-370: "In contrast, the most successful human-associated lineage Sm6 was linked to the presence of β -lactamases (BlaL1 and BlaL2) and aminoglycoside resistance-conferring enzymes (Aph) as well as KatA, involved in resistance to disinfectants, pointing towards adaptation to healthcare settings and survival on and in patients." I think this statement is a little misleading- by highlighting Sm6 in this way a reader may think that Sm6 is special in harbouring this list of genes, but most are present in a large proportion of the Sm complex. However, perhaps it is the combination of all of these that is particularly important in Sm6? Is it the only lineage with high prevalence of all of the named genes?

We added two sentences to ensure that readers are not misled in that Sm6 is the only lineage harbouring these genes: 'A striking finding was the observation that those lineages harbouring mostly environmental strains tended to harbour less resistance and virulence genes than lineages that comprised of the majority human-associated strains.' and 'While other human-associated lineages also harboured resistance and virulence genes at high proportions, this finding might explain why strains of lineage Sm6 were dominant in our investigation (...).'

12. Lines 373-374: "This notion is also supported by our finding that we did not detect any d100 clusters, or circulating variants, in the primarily environmental-associated lineages." The authors should mention here the potential role of differences in sample size and the lack of systematic samples for environmental isolates. Would we reasonably expect to find any groups of closely related clusters with the number of samples available? Particularly given that the numbers from any single location and time-frame will be low.

We added the following sentence at the end of this paragraph: 'Yet, in light of the low number of strains belonging to these lineages in our dataset as well as the lack of systematic environmental sampling, these results should be interpreted with caution.'

13. Line 398: "lineage Sm6 strains potentially best adapted to colonize or infect humans." – Typo- Colonise or infect humans

We refer to colonization when isolates were cultured from a patient without invasive infection, i.e. skin, wound, or perineal isolates. All isolates cultured from blood or CSF were considered human-invasive, and all isolates below the glottis from the respiratory tract as human-respiratory.

Methods:

14. Please specify how many SNPs were used for the BAPS and phylogenetic analyses i.e. the alignment length.

The total alignment length was 296,491 SNPs (invariant sites removed), we added this information to the Methods section in the respective paragraph.

15. Please expand the rationale for including urine isolates among the invasive infections rather than non-invasive? They are most commonly considered non-invasive. Could this have influenced the results and conclusions in any way?

Since *S. maltophilia* is not part of the normal flora of the human urinary tract, the clinicians in our consortium classified it as an infection (denoted as 'human invasive' in the manuscript) as opposed to isolates cultured from the skin, wounds, or the oral cavity (denoted as 'human-non-invasive'). Our collection comprises 36 urinary tract isolates that are distributed across several lineages, so we do not think that a different classification of these isolates would have influenced our results or conclusions. We will keep this in mind however, when we define our phenotype for follow up GWAS studies.

16. Please give more specific details about the methods for obtaining consensus allele calls e.g. coverage and identify cut-offs or k-mer depth thresholds.

The consensus calls are merely a combination of the assembly-based and assembly-free allele calling. The auto submission criteria, i.e. criteria that need to be met to get an allele number (existing or new), require a minimum homology with the reference allele > 80%, a maximum number of gaps of 999, the presence of a start/stop codon, and the absence of an internal stop codon. The parameters for assembly-based allele calling equally require minimum 80% similarity, gapped alignments are allowed, and a word size of 11. The parameters for assembly free calling are set at a kmer size of 35 with minimum coverage of 3. Only if both assembly free and assembly based call an allele (= consensus call) the respective allele will receive a number and be updated in the database.

We added this information in the methods section.

17. Please also give more details on the statement: "Biological replicates (sequencing data obtained from different fresh cultures of *S. maltophilia* strain ATCC 1363752) differed at maximum 5 consensus allele calls." This seems like a high number of differences for sequence data from recent subcultures of a single reference strain. What was causing the differences here? i.e. does this reflect a lack of allele calls in some genomes compared to others e.g. due to differences in sequencing depth? Or does this reflect allelic mismatches? The latter is more concerning as it calls into question the reproducibility of the results and is relevant to understanding the cluster analyses.

We investigated this validation analyses in more detail with our collaborators at bioMerieux. Since the technical replicates resulted in identical allele call profiles, we assume that varying allele calls on biological replicates are likely due to sequence variation that are below the threshold of receiving the exact allele call. Upon repeated review of these validation data, we observed that there is only one loci that differed between two samples (STENO00008), that was likely due to assembly errors in this high GC region.

There were 57 loci where one or more samples did not have an allele number while others had (i.e. no discrepant allele number but rather call or no call for these alleles). This could be caused by sequencing or assembly errors, or that some of the genes represented by the wgMLST loci are below the threshold to be called the same allele number. In this example, loci STENO00042, while one sample had a valid allele call, three samples had a sequence identity lower than the threshold. The freely and commercially available software tools that employ wgMLST schemes allow users to determine the thresholds (percentage similarity but also the size of indels), so there will be a tradeoff between maximum similarity to receive the same allele calls and the chance that assembly errors, mutations, and indels might led to the absence of an allele call if below the threshold.

One could certainly argue that such loci (i.e. representing genomic regions that are difficult to assemble) should be removed. We think, however, that in transmission analysis, which is the primary use of wgMLST in our view, this problem is only minor and does not affect our findings. If a few alleles are not called as they are below the similarity threshold, there will still be many other alleles that are called (~4000 allele calls, or depending on the genome size of the respective isolate).

We added a few sentences on this in the methods.

18. Please state the number of bootstrap replicates used in the phylogenetic analysis.

We used 100 bootstrap replicates. We added this information in the methods section in the corresponding paragraph.

19. Please indicate how many SNPs were masked from the alignment after ClonalFrame analysis.

The core gene alignment length was 1,070,730 variants, amounting to 1,397,302,650 characters for the entire dataset. Across all isolates, 593,506,119 positions (42% of all variants) were masked for recombination. For phylogenetic and BAPS analysis, all invariant sites were removed to obtain the final alignment length of 296,491 variants. We have added this information to the manuscript.

Other:

20. Figure 1a: I find the layout of the tree here a little confusing. It looks like the branches leading to Sgn4 and its sister clade have been truncated and moved to fit into the figure (?) but this is not obvious at first glance and not highlighted in the figure legend. I suggest rethinking the layout. Perhaps it would work to show a zoomed out view of the full tree (with all branches to scale and in true position) alongside a zoomed in view of just *S. maltophilia sensu lato*?

We changed the layout to show the full tree. The initial thought was to provide readers with more detail on the complex but one should be able to zoom into the figure. Having an inset with a zoomed in version of the sensu lato part would have taken too much figure space.

21. Fig 1d and 2a: Please consider reordering the lineage labels so that they are shown in numerical order (and coloured correspondingly).

We rearranged the lineage labels in Fig 1d, lineage labels in Fig 2a are already numbered in numerical order to align with those in Fig 2b and 2c. We left the lineage colours to ensure they are synchronised across the figures.

Reviewers' Comments:

Reviewer #1:

Remarks to the Author:

Congratulations to the entire team on carrying out an exhaustive genomic study on an emerging nosocomial pathogen that is also of biotechnological importance. Even though a major chunk of the novelty of the study is lost now as it is not a global study as declared in first version but still a large scale international study. Also the authors have addressed the other concerns and comments in the revised version. The unprecedented scale, plan and details of the study are of urgent need of the hour in our understanding of this superbug.

However I have one concern that is not clearly addressed and it is related to taking credit for proposing "Stenotrophomonas maltophilia complex". This terminology is used/stressed in several publications in past and more specifically in first large scale taxonogenomic study of the genus by Patil P and co-workers (ref 15). In fact any such proposal has been possible because of genomic resource of type of Stenotrophomonas made available from Patil P. study (ref 37). Even their newly provided Figure S5 by integrating data of ref 15, all the lineages fall between *S. rhizophila* and (further associated with) *S. maltophilia*. The finding is same as in as revealed in ref. 15 by Patil. P., 2018, Microbial Genomics. No other valid or historic species fall between sgn4 (strain 3123 in Patil P., et.al. ref) and *S. rhizophila*. Hence more than proposal it is strong support of "S. maltophilia complex" as revealed and highlighted by Patil, P and co-workers in the taxonogenomic study of the genus. Hence the authors need to cite earlier studies and clarify regarding "S. maltophilia complex" in introduction, results, and discussion.

Reviewer #2:

Remarks to the Author:

The revised version of the manuscript by Gröschel and colleagues has adequately addressed a number of concerns raised by the reviewers, with particular care and thought taken to address the concerns regarding the availability of the wgMLST scheme, the analysis of lineage-specific genes (which was removed and replaced by a formal association analysis) and the comments regarding taxonomic assignments. Regarding the latter, I fully understand that the authors do not wish to complicate the current manuscript by entering into a debate about taxonomic assignments; however I do think that it would help readers if in the section of the discussion describing the preferred nomenclature (lines 352-368), the authors note that the ANI values between some lineages are below the thresholds generally considered to define a species, and hence the taxonomic assignments and nomenclature for this group may be further revised in future studies. I think this point can be made without the need to propose novel species names and the associated biochemical descriptions etc. This article by Chun and colleagues (Int J Syst evol Micrbiol 2018: <https://www.ncbi.nlm.nih.gov/pubmed/29292687>) may be a useful reference for this point.

I have a small number of additional minor corrections and typographical errors that I feel the authors should address (see below), but no major concerns.

MINOR COMMENTS

Lines 175-177: "We calculated a phylogenetic tree based on 23 phylogenetic reference genes to visualize the clade formed by strains of the *S. maltophilia* complex within the genus *Stenotrophomonas* (Fig. S5)." The figure legend and methods section indicate that the phylogeny was inferred from an alignment of predicted amino acid sequences. Suggest indicating this here rather

than simply stating 'reference genes,' which could be taken to imply gene (nucleotide) sequences. There is also a duplicate period mark at the end of the sentence.

Lines 317-319: "When year and location of isolation were known and included in further investigations of the d10 clusters, we detected a total of 59 strains, grouped into 13 clusters, which were isolated from the same respective hospital in the same year." Please give an indication of these clusters in supplementary table S3 and indicate which strains belong to the same hospital. Currently there is information about country and city of isolation but there seem to be more than 13 clusters containing strains from the same city and year, so presumably some of these represent different hospitals in the same city?

Line 599-600 "based on an alignment of the concatenated protein sequence of 23 genes," Strictly speaking I think you should state "translated protein sequences" or "predicted protein sequences" or similar.

Line 613: "All assemblies were assessed for completeness and contamination using CheckM." I realise that this addition was a response to review but if this is stated in the text I think it should be followed by an indication of the range of values that were obtained. It should also be placed in the QC section rather than the phylogenetic section. (I note that I found the QC analysis in the original version of the manuscript and description thereof to be very thorough and that I agree with the authors that no further adjustments to the dataset were needed.)

Fig 5 and Fig S7: Please confirm that the cluster numbers described in the figures match those noted in the supplementary table S3 for the d10 column. None of the isolate names seems to match e.g. those shown as cluster 24 in the figure are marked as 42 in the table.

Fig S7: It's not clear why some of the clusters are shown using the core genes for the full collection whereas others are shown using the cluster specific core. Please clarify in the figure legend and preferably choose one approach to use for all.

TYPOS

Line 277 "phylogenetic lineage, ."

Lines 278 "in at least 10 isolates were selected"

Lines 387-389: "independent events of pathoadaptation of environmental strains to human colonization, as has been observed for *Legionella pneumophila* 40, , or . A more recent study expanded these findings on the entire *Legionella* genus,"

Line 385: "colonization and infection. almost all lineages contain isolates"

Line 554: "a lastn search against assembled genomes"

Line 577: "the allelic profiles differed as follows i ."

Line 581 "loci STENO00042" - loci should be locus?

REVIEWERS' COMMENTS:

Reviewer #1 (Remarks to the Author):

Congratulations to the entire team on carrying out an exhaustive genomic study on an emerging nosocomial pathogen that is also of biotechnological importance. Even though a major chunk of the novelty of the study is lost now as it is not a global study as declared in first version but still a large scale international study. Also the authors have addressed the other concerns and comments in the revised version. The unprecedented scale, plan and details of the study are of urgent need of the hour in our understanding of this superbug.

However I have one concern that is not clearly addressed and it is related to taking credit for proposing "Stenotrophomonas maltophilia complex". This terminology is used/stressed in several publications in past and more specifically in first large scale taxonogenomic study of the genus by Patil P and co-workers (ref 15). In fact any such proposal has been possible because of genomic resource of type of Stenotrophomonas made available from Patil P. study (ref 37). Even their newly provided Figure S5 by integrating data of ref 15, all the lineages fall between *S. rhizophila* and (further associated with) *S. maltophilia*. The finding is same as in as revealed in ref. 15 by Patil. P., 2018, Microbial Genomics. No other valid or historic species fall between sgn4 (strain 3123 in Patil P., et.al. ref) and *S. rhizophila*. Hence more than proposal it is strong support of "S. maltophilia complex" as revealed and highlighted by Patil, P and co-workers in the taxonogenomic study of the genus. Hence the authors need to cite earlier studies and clarify regarding "S. maltophilia complex" in introduction, results, and discussion.

We thank the reviewer both for considering our revised manuscript and the positive comments regarding its importance for the field. As pointed out by the reviewer, we do indeed build upon previous work, and greatly appreciate the work and contributions that Patel P and colleagues have made in the field. Following the reviewer's advice, we carefully went over the manuscript to ensure citations of previous works are included wherever applicable and to rephrase the text where necessary to clearly indicate previous reports.

We introduce and define the *S. maltophilia* complex in the second paragraph of the introduction and cite 4 relevant publications, including Patil P. *et al* 2018 (ref 15):

"Previous work indicated the presence of at least 13 lineages or species-like lineages in the ***S. maltophilia* complex, defined** as *S. maltophilia* strains with 16S rRNA gene sequence **similarities > 99.0%**, with nine of these potentially human-associated¹⁴⁻¹⁸."

To ensure that readers do not conclude that we propose the term *S. maltophilia* complex for the first time, we added the wording in the discussion: "We therefore propose to *continue* using the term *S. maltophilia* complex ...". The publications proposing and defining the *S. maltophilia* complex are also cited in the introduction and results section.

As we discuss the lineages, we point to earlier studies that first described *sensu lato* and *sensu stricto*:

"In concordance with these studies^{14,16}, we found a clear separation of the more distantly related lineages Sgn1-Sgn4 and a branch formed by lineages Sm1-Sm18 (previously termed *S. maltophilia sensu lato*), with the largest lineage Sm6 (also known as *S. maltophilia sensu stricto*)".

Reviewer #2 (Remarks to the Author):

The revised version of the manuscript by Gröschel and colleagues has adequately addressed a number of concerns raised by the reviewers, with particular care and thought taken to address the concerns regarding the availability of the wgMLST scheme, the analysis of lineage-specific genes (which was removed and replaced by a formal association analysis) and the comments regarding taxonomic assignments. Regarding the latter, I fully understand that the authors do not wish to complicate the current manuscript by entering into a debate about taxonomic assignments; however I do think that it would help readers if in the section of the discussion describing the preferred nomenclature (lines 352-368), the authors note that the ANI values between some lineages are below the thresholds generally considered to define a species, and hence the taxonomic assignments and nomenclature for this group may be further revised in future studies. I think this point can be made without the need to propose novel species names and the associated biochemical descriptions etc. This article by Chun and colleagues (Int J Syst evol Microbiol 2018: <https://www.ncbi.nlm.nih.gov/pubmed/29292687>) may be a useful reference for this point.

I have a small number of additional minor corrections and typographical errors that I feel the authors should address (see below), but no major concerns.

We wish to thank the reviewer for examining our revised manuscript and are glad to hear that we were able to address the reviewer's concerns. We are also thankful for the suggestion to include a sentence on further work on the taxonomic assignments which we added in the discussion accordingly.

MINOR COMMENTS

Lines 175-177: "We calculated a phylogenetic tree based on 23 phylogenetic reference genes to visualize the clade formed by strains of the *S. maltophilia* complex within the genus *Stenotrophomonas* (Fig. S5)." The figure legend and methods section indicate that the phylogeny was inferred from an alignment of predicted amino acid sequences. Suggest indicating this here rather than simply stating 'reference genes,' which could be taken to imply gene (nucleotide) sequences. There is also a duplicate period mark at the end of the sentence.

We added this information to this sentence. We removed the double period at the end of the sentence.

Lines 317-319: "When year and location of isolation were known and included in further investigations of the d10 clusters, we detected a total of 59 strains, grouped into 13 clusters, which were isolated from the same respective hospital in the same year." Please give an indication of these clusters in supplementary table S3 and indicate which strains belong to the same hospital. Currently there is information about country and city of isolation but there seem to be more than 13 clusters containing strains from the same city and year, so presumably some of these represent different hospitals in the same city?

We updated the figures and text by the original cluster numbers so that they match the supplementary data. We also add in the text that we define clusters as having at least 2 isolates, and that we took the isolation date rather than year to filter down to the 13 clusters. We hope that it is now much clearer to the reader. We have noted the hospitals (Hospital A, etc) behind the respective cluster numbers in the supplementary file.

Fig 5 and Fig S7: Please confirm that the cluster numbers described in the figures match those noted in the supplementary table S3 for the d10 column. None of the isolate names seems to match e.g. those shown as cluster 24 in the figure are marked as 42 in the table.

We have updated these figures with the cluster numbers listed in the supplementary data.

Line 599-600 “based on an alignment of the concatenated protein sequence of 23 genes,” Strictly speaking I think you should state “translated protein sequences” or “predicted protein sequences” or similar.

We added 'predicted' before 'protein sequences', as suggested.

Line 613: “All assemblies were assessed for completeness and contamination using CheckM.” I realise that this addition was a response to review but if this is stated in the text I think it should be followed by an indication of the range of values that were obtained. It should also be placed in the QC section rather than the phylogenetic section. (I note that I found the QC analysis in the original version of the manuscript and description thereof to be very thorough and that I agree with the authors that no further adjustments to the dataset were needed.)

We moved this information to the QC section and added the ranges, mean and SD as suggested.

Fig S7: It's not clear why some of the clusters are shown using the core genes for the full collection whereas others are shown using the cluster specific core. Please clarify in the figure legend and preferably choose one approach to use for all.

Our aim was to compare the general core genome MLST of the study collection versus the cluster specific core MLST with regards to their outbreak resolution, and showing that it's useful to work with outbreak specific core genome MLST schemes, which can be easily done and is - in our view - one of the advantages of wgMLST schemes. We added one sentence to make sure this is clear to the reader.

TYPOS

Line 277 “phylogenetic lineage, .”

Lines 278 “in at least 10 isolateswere selected”

Lines 387-389: “independent events of pathoadaptation of environmental strains to human colonization, as has been observed for Legionella pneumophila40. , or . A more recent study expanded these findings on the entire Legionella genus,”

Line 385: “colonization and infection.salmost all lineages contain isolates” Line 554: “a lastn search against assembled genomes”

Line 577: “the allelic profiles differed as followsi .”

Line 581” “loci STENO00042” - loci should be locus?

We thank reviewer #2 for spotting these typos which we corrected in the manuscript.